# Widespread dissemination of *Salmonella, Escherichia coli* and *Campylobacter* resistant to medically important antimicrobials in the poultry production continuum in Canada

Hiddecel Medrano[1], Sarah Hill[2,3], Martine Boulianne[4], Teresa Cereno[5], Anne E. Deckert[2,3], Audrey Charlebois[6], Sheryl P. Gow[7], Kathryn McDonald[1], Richard J. Reid-Smith[2], Agnes Agunos[2]*

1 Public Health Agency of Canada (PHAC), Calgary, Alberta, Canada, 2 PHAC, Guelph, Ontario, Canada, 3 Department of Population Medicine, University of Guelph, Guelph, Ontario, Canada, 4 Chair in Poultry Research, Faculté de médecine vétérinaire, Université de Montréal, St-Hyacinthe, Québec, Canada, 5 Canadian Food Inspection Agency, Ottawa, Ontario, Canada, 6 National Microbiology Laboratory, PHAC, St-Hyacinthe, Québec, Canada, 7 PHAC, Western Veterinary College, Saskatoon, Saskatchewan, Canada

* agnes.agunos@phac-aspc.gc.ca

## Abstract

The Canadian Integrated Program for Antimicrobial Resistance Surveillance (CIPARS) monitors *Escherichia coli, Salmonella* and *Campylobacter* and their resistance to antimicrobials in broiler chickens at the farm and slaughter plant levels. In response to many years of CIPARS' observations and farmers' data, the Chicken Farmers of Canada implemented a strategy to reduce antimicrobial use in 2014. As resistance genes can be transmitted vertically from parents to their offspring, a study was conducted in broiler breeder flocks to assess the frequency of target bacteria, their antimicrobial resistance (AMR) and to obtain a comprehensive picture of AMR in poultry production. Spent breeder flocks slaughtered between 2018 and 2021 were sampled and data from broiler flocks at the farm and slaughter plants were assessed. *Salmonella* was most frequently detected in farm broiler chickens (46%), while *Campylobacter* was most frequently detected in broiler breeders (73%). In *Campylobacter*, high levels (20–24%) of ciprofloxacin resistance were found across the three production stages, and was highest in farm broiler chickens (24%). In *E. coli,* an indicator organism, low-level ceftriaxone resistance and occasional isolates that were non-susceptible to ciprofloxacin were noted. Using the indicator, fully susceptible *E. coli*, broiler breeders had the highest frequency (54%) compared to farm (36%) and slaughtered (35%) broiler chickens. In *Salmonella* broiler breeders had the highest resistance to most antimicrobials tested. Fully susceptible *Salmonella* was lowest in broiler breeders (16%) compared to farm (42%) and slaughtered (42%) broiler chickens. *Salmonella* serovars differed between the production stages, but *S.* Kentucky was the most predominant. Resistance to critically important antimicrobials in human medicine and regional variations in resistance profiles were observed. This study suggests that broiler breeders carry foodborne bacteria resistant to antimicrobials used in human medicine, demonstrating their role in the maintenance of AMR in poultry and the need to adopt a harmonized sector-wide AMU strategy.

**Data availability statement:** All relevant data are within the manuscript and are available via the Public Health Agency of Canada's data repository and interactive data visualization platform: https://www.canada.ca/en/public-health/services/surveillance/canadian-integrated-program-antimicrobial-resistance-surveillance-cipars/interactive-data.html

**Funding:** Canadian Poultry Research Council.

**Competing interests:** The authors have declared that no competing interests exist.

## Introduction

According to the World Health Organization (WHO), the responsible and prudent use of antimicrobials needs to be improved to preserve their benefits [1]. Antimicrobial resistance (AMR) is a global problem affecting human, environmental, plant/crop and animal health. Contributing to this complex issue, are not only the use of antimicrobials in all of these sectors but also the rapid spread of antimicrobial resistance genes across them.

The Canadian Integrated Program for Antimicrobial Resistance Surveillance (CIPARS), administered by the Public Health Agency of Canada (PHAC), has been monitoring AMR in food animals since 2002 [2] through the surveillance of foodborne bacteria in different sample matrices and from various collection points throughout the food chain [3]. Results demonstrate the widespread dissemination of *Salmonella* and *Campylobacter* in Canadian chickens and their products [4]. Both bacteria are the leading cause of illness annually in Canada [5], with an annual incidence rate of 12.4 per 100,000 population for *Salmonella* spp. cases and 2.8 per 100,000 population for *Campylobacter* spp. cases in 2022 [6]. Infection with antimicrobial-resistant bacteria may pose an additional public health burden when antimicrobials are no longer effective in treating the disease.

Routine antimicrobial susceptibility testing (AST) of these isolates indicates that AMR in foodborne bacteria persists in Canadian chickens [4]. Observed resistance to antimicrobial classes deemed as highest priority critically-important antimicrobials (HP-CIA's) by the World Health Organization [1] and very high importance according to Health Canada's Veterinary Drugs Directorate (Category I antimicrobials) [7], such as 3$^{rd}$ generation cephalosporins in generic *Escherichia coli* and *Salmonella* and fluoroquinolones in *Campylobacter,* are of concern [4].

To mitigate AMR in the poultry sector, the Chicken Farmers of Canada adopted a strategy that saw the formal elimination of the preventive use of Category I antimicrobial classes, cephalosporins and fluoroquinolones in 2014, and high importance antimicrobials (Category II antimicrobials), such as aminoglycosides, macrolides, lincosamides, penicillins, streptogramins, trimethoprim/sulfamethoxazole, at the end of 2018 [8]. Information on the prevalence of foodborne *Salmonell*a and *Campylobacter* and their levels of resistance to antimicrobials at relevant poultry production stages (food production to consumption continuum starting from the breeder stage to slaughter/point of sale), is useful in contextualizing AMR food safety issues from poultry products [9], specifically in meat-type broiler chicken commodity. Collected data might also help to validate if AMU reduction strategies in the livestock industry are an effective intervention. Depending on the bacterial species' transmission dynamics, for example, in *Salmonella*, the upper level of the food production chain (broiler breeders or parent stock raised for the production of broiler meats) can play a role in the dissemination of *Salmonella*/antimicrobial-resistant *Salmonella* to other production stages. As such, monitoring of foodborne bacteria such as *Salmonella* and *Campylobacter*, along with AMR indicator bacteria (*Escherichia coli*) inform the design of interventions to control their potential food safety and public health implications [9]. In Canada, surveillance of antimicrobial use (AMU) and AMR in sentinel broiler farms, and AMR in broiler chickens at the slaughter plant level are conducted as part of CIPARS. However, a knowledge gap persists on the AMU/AMR situation in broiler breeders in Canada.

Research studies have found *Salmonella* in broiler breeders [10–12]. The dissemination of this organism from the broiler breeder flocks to their offspring/hatched chicks [11] has been described, emphasizing the role of broiler breeders in the maintenance and dissemination of *Salmonella*/antimicrobial-resistant *Salmonella* in the food production to consumption continuum. In Ontario, one of the major poultry-producing provinces in Canada, two studies

were conducted to determine the prevalence of *Salmonella* from breeder flock environment, the first study (sampling years: 1998–2008) found 47.4% positive samples [10] and in the second study (sampling years: 2009–2018) found 25.4% positive samples [13]. The latter study detected clusters of serovars from the broiler breeder environment which were similar to the serovar clusters identified in hatchery fluff samples (feathers shed by hatching chicks collected at the hatcher machine), further demonstrating the vertical transmission or cross-contamination of *Salmonella* from breeder level to the next stages of production (incubation at the hatchery and chicks placed in broiler farms for chicken meat production). At the hatchery, *Salmonella*-positive hatching eggs that were sourced from *Salmonella*-positive broiler breeders could become the source of cross-contamination between chicks during the hatching process [14]. *Salmonella*-positive chicks could subsequently contaminate the broiler barn environment [15].

Other than *Salmonella*, the transmission of extended spectrum cephalosporinase (ESC)-producing *E. coli,* including potentially human pathogenic types, from broiler breeders to their offspring has been described [16]. A Canadian study comparing samples collected in breeders and their progeny, pre- and post-voluntary elimination of preventive uses of Category I antimicrobials demonstrated a decrease in ceftiofur resistance genes ($bla_{CMY-2}$ and $bla_{CTX-M}$) one year after the cessation, although the resistance genes remained present in all phases of poultry production [17]. Vertical transmission of resistance genes was pinpointed as a likely source. The vertical transfer of $bla_{CTX-M-1}$-harboring Incl1 plasmids in *E. coli* with different Multilocus Sequence Types (MLST) within the poultry production continuum has been reported [18], while another study showed vertical transfer of Incl1 and IncK plasmids carrying $bla_{CTX-M-2}$ in *E. coli* [19]. A case of fluoroquinolone-resistant *E. coli* transferred from breeders to their progenies has also been described in Denmark where fluoroquinolone use in poultry is restricted [20].

*Campylobacter* is another foodborne pathogen of concern in the poultry industry. However, the exact role of the breeder flocks in the emergence of resistant strains and their dissemination in the poultry industry, whether through vertical transfer from breeders to progeny flocks, horizontally through cross-contamination between production facilities, or both, has not yet been established [21]. Of important public health concern are the stable levels of resistance to ciprofloxacin, a fluoroquinolone antimicrobial categorized as very high importance and highest priority critically important to human medicine by the Veterinary Drugs Directorate, Health Canada and the World Health Organization, respectively [7,22].

In terms of antimicrobial exposures, CIPARS has been collecting AMU data from sentinel broiler chicken farms since 2013. While the withdrawal of ceftiofur at hatcheries mirrored a decreasing resistance to 3rd generation cephalosporins [23], it also prompted an increase in the use of and resistance levels for other drug classes, such as aminoglycosides [24]. More precisely, Verrette et al. [17], showed that the replacement of ceftiofur by lincomycin-spectinomycin led to an increase in multidrug resistance with profiles containing resistance to aminoglycosides, folate pathway inhibitors (trimethoprim and sulfonamides), phenicols, and tetracyclines and a greater proportion of possible extensively drug-resistant *E. coli.* More recently the preventive use of lincomycin-spectinomycin (a Category II antimicrobial) has also been eliminated and the trends in the prevalence of multidrug-resistant (MDR) *E. coli* and MDR *Salmonella* were similar to the trends in total AMU measured in number of defined daily doses using Canadian standards (nDDDvetCA) per 1,000 broiler chicken-days at risk [25]. As previously described, CIPARS does not conduct surveillance of AMU in broiler breeder flocks, and the levels of exposure to antimicrobials and their impact on AMR are unknown.

To understand foodborne pathogen status and AMR in the food production to fork continuum in broiler chickens, a cross-sectional study was conducted to evaluate the national

prevalence of AMR in *Salmonella, Campylobacter,* and generic *E. coli* in healthy slaughtered spent broiler breeders (end of egg laying period) and the AMR in broiler chickens at the farm and slaughter plant level between 2018 and 2021. This study aims to address the knowledge gaps on the role of broiler breeders in the ecology of AMR in broiler chickens in Canada and to contribute data for foodborne AMR risk profiling studies and inform industry-wide interventions to limit the development and spread of AMR across the broiler food production to fork continuum.

## Methods

### Sample collection

**Collection timeframe.** Broiler breeders were sampled from June 2018 through December 2021. In two provinces (Québec and Manitoba), a total of 17 flocks actively producing hatching eggs for broiler chicken production in 2021, but only available for sampling in early 2022 were included in the study to enhance data representativeness. The AST data routinely collected by CIPARS from broilers at the farm and slaughter plants corresponding to the breeder flock sample collection timeframe were included in the analysis [3].

**Broiler breeders.** The minimum required total breeder flocks for this study (n = 290 flocks) were distributed across the voluntary participating plants (5 plants across the country) based on their projected slaughter volume for spent breeders. Five to 6 cecal samples were collected at the slaughter plant for each broiler breeder flock (one farm source) sent for slaughter at the end of the egg-laying period [3]. The breeder flock record ("flock sheet") was also obtained if available. For each participating slaughter plant, breeder flocks slaughtered during the sampling collection timeframe were sampled.

**Farm broiler chickens.** Sentinel broiler flocks (on average, n = 132 flocks per year across five provinces) were visited during the last week of growth (> 30 days of age), once per year per farm for sample and antimicrobial use (AMU) data collection. Four pooled fecal samples, representing 1 per floor quadrant with at least 10 fresh fecal droppings (60 grams total quantity) were collected from randomly selected barns [3].

**Slaughtered broiler chickens.** Samples were obtained from federally inspected plants (on average, n = 24 slaughter plants) and the frequency of sampling and total samples collected were based on the slaughter volumes. During each sampling week, seven caecal samples were collected within five days, at the convenience of the slaughterhouse staff, provided the animals and associated samples originated from different groups. Sampling from different groups of animals is important to maximize diversity and avoid bias attributable to overrepresentation of particular producers. Collection periods are uniformly distributed throughout the year to avoid any bias that may result from seasonal variation in bacterial prevalence and AST results [3].

### Bacterial isolation and characterization

For sample preparation, 25 grams of each fecal sample from broiler chickens sampled at the farm were mixed with 225 mL of buffered peptone water (BPW). Caecal samples from broiler breeders and broiler chickens at the slaughter plant were weighed and mixed with BPW at a 1:10 ratio, then incubated at 35 ± 1 ºC for 18–24 hours. For *E. coli* isolation, 0.1 mL of the BPW mixture was streaked onto MacConkey agar and incubated at 35 ± 1 ºC for 18 to 24 hours. Suspect lactose-fermenting colonies were screened for purity and transferred onto Luria-Bertani agar. Presumptive *E. coli* colonies were assessed using Simmons citrate and indole tests. The isolates with negative indole test results were confirmed using a bacterial identification test kit (API-20E) [3].

For *Salmonella* isolation, 0.1 mL of the BPW mixture previously prepared was used to inoculate a modified semi-solid Rappaport-Vassiliadis (MSRV) plate and incubated for 42 ± 1 ºC for 24 to 72 h. Isolates with greater than or equal to 20 mm migration were streaked onto MacConkey agar and incubated at 35 ± 1 ºC for 18 to 24 hours. Suspect colonies (lactose-negative) were screened for purity and used to inoculate triple-sugar-iron and urea agar slants. Presumptive *Salmonella* isolates were confirmed using the indole test, and were identified via slide agglutination with *Salmonella* Poly A-I and Vi antiserum. The serotyping of *Salmonella* isolates was done through the traditional phenotypic serotyping or *Salmonella* GenoSero-typing Array (SGSA) until May 2019. In June 2019, isolates were sequenced and typed using *Salmonella in silico* Typing Resource (SISTR). The phenotypic serotyping method detects O or somatic antigens of the *Salmonella* isolates through slide agglutination, while the SGSA detects the genes encoding the O and H antigens. SISTR detects the genes in the genome sequence which encodes the surface O and H antigens. All methods report the corresponding *Salmonella* serovar according to the White-Kauffmann-Le Minor (WKL) scheme [3].

For *Campylobacter* isolation, 1 mL of the BPW mixture prepared previously was mixed with 9 mL of Hunt's enrichment broth (HEB), then incubated in a microaerophilic atmosphere at 35 ± 1 ºC for 4 h. Next, 36 μL of sterile cefoperazone were added to the HEB tubes, and incubated at 42 ± 1 ºC for 20 to 24 h. The HEB mixture was used to saturate a swab which was then used to inoculate a modified Charcoal Cefoperazone Deoxycholate Agar (mCCDA) plate. This plate was incubated at 42 ± 1 ºC in microaerophilic conditions for 24 to 72 h. Suspect colonies were streaked onto a second mCCDA and incubated. From this culture step, a colony was streaked onto a Mueller Hinton with sheep blood agar plate. This was then incubated at 42 ± 1 ºC in a microaerophilic atmosphere for 24 to 48 h. Gram stain, oxidase and catalase tests were used to identify presumptive colonies. Mutiplex PCR (mPCR) was used to determine the species of *Campylobacter* in selected colonies, where specific genomic targets (hippuricase in *C. jejuni* and aspartokinase in *C. coli*) were amplified from bacterial lysates [3].

## Antimicrobial susceptibility testing

All bacterial isolates (*E. coli, Salmonella* and *Campylobacter*) from farm broiler chickens and slaughtered broiler breeders were tested. However, in slaughtered broiler chickens, only one in every four *E. coli* isolates were randomly selected for AST.

Sensititre plates designed by the United States National Antimicrobial Resistance Monitoring Program (US NARMS) were used for AST. For Gram-negative bacteria, each plate consisted of 14 antimicrobials, and different plate configurations were available in 2018–2019 (CMV4AGNF) and 2020–2022 (CMV5AGNF) (Sensititre, Trek Diagnostics Systems, West Sussex, England). Colistin was added to the study panel in 2020. For *Campylobacter*, each plate consisted of 9 antimicrobials and different plate configurations were available in 2018–2019 (CAMPY) and 2020–2022 (CMVCAMPY) (Sensititre). The Clinical Laboratory Standards Institute (CLSI) interpretive criteria, where available, were used to interpret the minimum inhibitory concentrations (MICs) [26–28]. In cases where CLSI breakpoints were unavailable, breakpoints based on the US NARMS [29] and CIPARS clinical breakpoints [30] were used.

## Data Analysis

The recovery of generic *E. coli, Salmonella* and *Campylobacter* across the production phases were recorded. STATA SE/V17 software (Statacorp, College Station, TX, USA) was used for all analyses including the proportions of bacteria presented at the sample level, descriptive statistics, and categorization of AMR unless otherwise stated. Using the Health Canada's

VDD categorization [7], antimicrobials were classified from I through IV based on their importance in human medicine in the Canadian context. Antimicrobials in Category I are identified as those of the highest importance in human medicine, while Categories II through IV represent decreasing importance. The antimicrobials used in this analysis were: Category I: amoxicillin-clavulanic acid, colistin, ciprofloxacin, ceftriaxone, and meropenem. Category II: ampicillin, azithromycin, cefoxitin, clindamycin, erythromycin, gentamicin, nalidixic acid, and trimethoprim-sulfamethoxazole. Category III: chloramphenicol, florfenicol, sulfisoxazole, and tetracycline. Antimicrobial susceptibility results are presented at the isolate level and were classified as susceptible, intermediate, or resistant using the CLSI interpretive criteria. Results were further dichotomized into binary outcomes (resistant = 1, susceptible and intermediate = 0) for all antimicrobials except ciprofloxacin where *E. coli* and *Salmonella* isolates classified as resistant and intermediate based on the CLSI interpretative criteria were combined and reported as non-susceptible isolates. We characterized the non-susceptibility of ciprofloxacin *E. coli* and *Salmonella* isolates as with the US NARMS where intermediate and resistant isolates according to the CLSI ($\geq 0.12$ μg/mL) are reported as isolates with decreased susceptibility to ciprofloxacin. The change in breakpoints is reflective of the whole genome sequencing for AMR determination in Enterobacterales, specifically the detection of mutation in *gyrA* in the isolates [3]. Isolates resistant to at least three antimicrobials are referred to as MDR isolates, as described by the European Food Safety Authority (EFSA) [31] and Magiorakos et al. [32].

The descriptive statistics and frequencies of resistance were reported as percentage of isolates resistant and 95% confidence intervals (CI) were determined. The MIC's were visually inspected and $MIC_{50}$ (minimum dilution where at least 50% of the isolates were inhibited) and $MIC_{90}$ (minimum dilution where at least 90% of the isolates were inhibited) were determined. For the description of the percentage of resistance, the EFSA's methodology [33], were used, from rare [$< 0.1\%$ of isolates] to extremely high level [70% isolates].

Univariable logistic regression analyses were performed to measure the association between studies (production phase) and AMR within each study. Variance-covariance matrix (VCE) was used to adjust for clustering at the isolate level. Conforming to routine CIPARS reporting of AMR indicators with potentially high public health significance, resistance to specific Category I antimicrobials and extremely high levels of AMR (tetracycline) were the primary focus of this paper. All statistical significance was defined by *p* values $\leq 0.05$. Additional AMR outcomes presented are the number of antimicrobial classes each isolate was resistant to, and the distribution of the number of classes to which an isolate was resistant to. AMR phenotypes were used to illustrate resistance distributions in heatmaps in RStudio's 'Gplots' R package (Version 2023.12.0 'Ocean Storm' [34]. Other RStudio packages used include 'Tidyverse' [35], 'RcolorBrewer' [36], and 'Readr' [37]. The heatmaps were used to visualize the hierarchical clustering of the microorganisms and *S.* Kentucky which was the most common across the country and production stages and where resistance to antimicrobials was expressed, where the y-axis (right) corresponded to the isolate and re-ordered based on hierarchal clustering. The y-axis on the left, and z-axis on the top displays the distance of similarity visualized as dendrogram. The colours of the map represented whether the isolate was resistant or susceptible to the particular antimicrobial in that column.

## Farmer informed consent

The data used for this study were extracted from the CIPARS AMR data repositories and spreadsheets. Except for the broiler breeder isolates, all samples were collected as part of CIPARS' routine systematic sampling in healthy terrestrial animal populations. For the farm

component, samples were taken from the immediate environment of the sentinel flocks, whereas, for the slaughter plant component (slaughtered broiler chickens and spent broiler breeders) samples were taken after the evisceration stage. The study did not involve animal experiments and direct contact with live birds was not necessary, thus no ethics approval was required. The farm surveillance protocols were reviewed by the poultry industry AMU/ AMR ad hoc working group during the development of the farm-level surveillance framework between 2009 and 2011. The sentinel poultry veterinarian administered an informed consent form to the farmer before accessing the farm records and entering the barn for data and sample collection, respectively. The cecal samples from broiler chickens and spent broiler breeders were taken from federally inspected slaughter plants that volunteered to participate in CIPARS by providing cecal samples.

## Results

### E. coli, Salmonella, and Campylobacter recovery

As shown in Table 1, generic *E. coli* was recovered from between 95.6% and 97.4% of the samples across the three production stages, where isolates were collected from broiler breeders (n = 1,555 samples), farm broiler chickens (n = 2, 025), and slaughtered broiler chickens (n = 1,178). The percentage of *Salmonella*-positive samples varied across the production phases with the lowest recovery rate at 10.7% in broiler breeders and the highest rate at 46.4% in farm broilers. There were 168 *Salmonella* isolates recovered from broiler breeders, while 977 and 489 isolates were recovered from farm and slaughtered broiler chickens respectively. *Campylobacter*-positive samples varied across production phases, from 22.3% recovery in slaughtered broiler chickens to 73.2% recovery in broiler breeders. With 1,168 isolates, broiler breeders had the highest number of isolates collected comprising 470 (40.2%) *C. jejuni*, 665 (56.9%) *C. coli*, and 33 (2.8%) unspeciated *Campylobacter* isolates (*Campylobacter* spp.). This is followed by slaughtered and farm broiler chickens with 677 (*C. jejuni*: 581 (85.8%); *C. coli*: 91 (13.4%); *Campylobacter* spp.: 5(7.4%)) and 465 (*C. jejuni*: 412 (88.6%); *C. coli*: 53 (11.4%)) isolates collected, respectively. Taken together, *C. jejuni* was the predominant *Campylobacter* species found across the three stages of the poultry food production continuum under surveillance, although the proportion of *C. coli* was substantially higher in broiler breeders than in broilers at the farm and slaughter plant levels.

### Escherichia coli

**Resistance percentage.** For antimicrobials listed in VDD Category I, no resistance to colistin or meropenem was detected in *E. coli* across all stages of production. Very low to

**Table 1. Proportions of *E. coli, Salmonella,* and *Campylobacter* positive samples between 2018–2021, by production phase.**

| Production stages | Sampling location | Proportion of positive samples | | |
| --- | --- | --- | --- | --- |
| | | *E. coli* % (n/N) | *Salmonella* % (n/N) | *Campylobacter* % (n/N) |
| Broiler breeders* | Slaughter plant | 97.4 (1,555/1,596) | 10.7 (168/1,596) | 73.2 (1,168/1,596) |
| Farm broiler chickens | Farm | 96.2 (2,025/2,104) | 46.4 (977/2,104) | 22.7 (465/2,060)[a] |
| Slaughtered broiler chickens | Slaughter plant[b] | 95.6 (1,178/1,232) | 15.9 (489/3,076) | 22.3 (677/3,060) |

*Broiler breeders include (n = 1,596) samples taken from early 2022 (these birds produced the vast majority of their hatching eggs in 2021).

[a]Forty-four samples were not tested in 2020.

[b]A total of 3,089 samples were submitted during the study timeframe. Final sample size varied depending on the organism: *E. coli* (1 in 4 samples were tested), *Salmonella* (13 samples were unfit for testing or lost specimen) and *Campylobacter* (24 samples unfit for testing or lost specimen).

low-level resistance to ceftriaxone was observed across production stages, but it was less pronounced in broiler breeders. Low levels of ciprofloxacin non-susceptible isolates were detected across all production stages. Non-susceptible ciprofloxacin isolates collected from farm and slaughtered broiler chickens were significantly higher ($p < 0.001$) than in broiler breeders. The highest levels were found in slaughtered broiler chickens at 9.7%, 95% CI: 8.0–11.5%, followed by farm broiler chickens and broiler breeders (8.3%, 95% CI: 7.1–9.5% and 2.57%, 95% CI: 1.8–3.5% respectively). The ciprofloxacin non-susceptible isolates paralleled the levels of nalidixic acid resistance. For the remaining antimicrobials from across all production stages, there was very low to low-level resistance to amoxicillin-clavulanic acid (2.4–6.1%, 95% CI: 1.7–7.1%) and moderate to high levels of resistance to ampicillin (20.3–32.1%, 95% CI: 18.3–34.2%), sulfisoxazole (10.7–40.1%, 95% CI: 9.0–43.9%) and tetracycline (34.2–37.5%, 95% CI: 31.8–39.6%) (Fig 1). Taken together, resistance levels at the broiler breeder level were lowest for 12 antimicrobials compared to farm broiler and slaughtered broiler chicken isolates. Of the 14 antimicrobials on the panel, non-significant differences between production stages were observed only for three antimicrobials (azithromycin, trimethoprim-sulfamethoxazole and tetracycline).

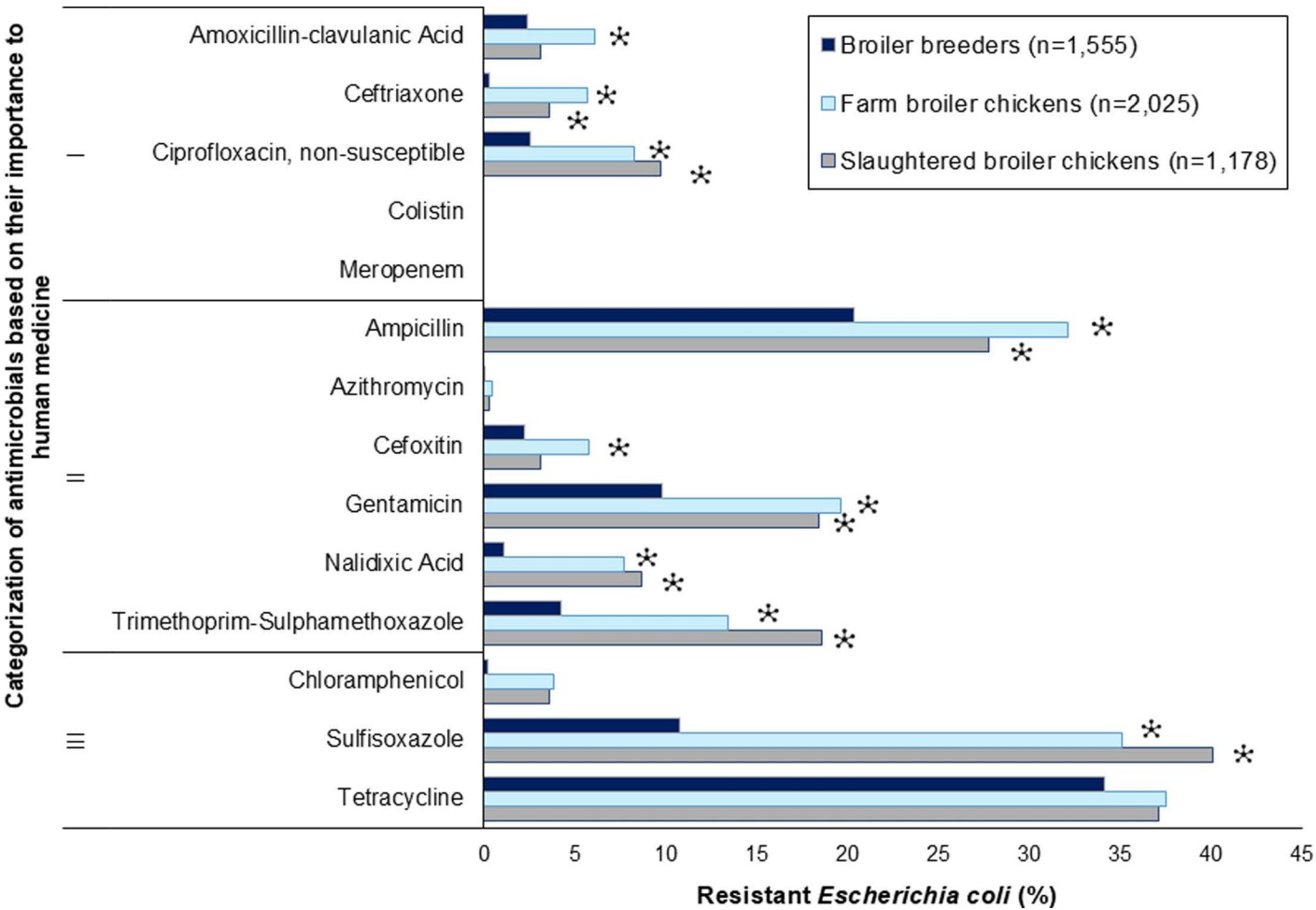

**Fig 1. Percentage of resistance in *E. coli* from broiler breeders, farm broiler chickens, and slaughtered broiler chickens, 2018–2021.** * significantly higher vs. broiler breeders (the referent) ($p = \leq 0.05$).

## Multidrug resistance, AMR patterns, heatmap resistance distribution

More than half of the broiler breeder isolates (54.5%, 95% CI: 52.0–57.0%) were fully susceptible to all antimicrobials in the AST panel, while 36.4% (95% CI: 34.3–38.6%) and 35.1% (95% CI: 32.2–37.9%) of all farm and slaughtered broiler chicken isolates respectively were fully susceptible. Of the broiler breeder isolates, 7.5% (95% CI: 6.3–8.9%) were resistant to three or more antimicrobial classes (MDR) while 22.1% (95% CI: 20.3–24.0) and 21.8% (95% CI: 19.5–24.3%) of farm and slaughtered broiler chicken isolates, respectively, were MDR (Table 2).

The most common phenotypic pattern in generic *E. coli* isolates was fully susceptible. Other patterns observed were tetracycline resistance (broiler breeders: 16.5% (95% CI: 14.7–18.5%)), farm broiler chickens: 7.9% (95% CI: 6.7–9.1%); slaughtered broiler chickens: 6.8% (95% CI: 5.4–8.4%%], ampicillin-tetracycline resistance in broiler breeders (7.3% (95% CI: 6.0–8.7%)) and ampicillin resistance in farm and slaughtered broiler chickens (5.8% (95% CI: 4.8–6.9%) and 5.0% (95% CI: 4.7–7.4) respectively).

Fig 2 illustrates a heatmap of how *Escherichia coli* isolates based on their resistance profiles could be related. The heatmap depicts the predominance of *E. coli* isolates resistant to tetracycline across all production stages, tetracycline-ampicillin in broiler breeders, and tetracycline-sulfisoxazole in farm and slaughtered broiler chickens. All predominant

**Table 2. Multidrug resistance in *E. coli, Salmonella,* and *Campylobacter* in broiler breeders, farm broiler chickens, and slaughtered broiler chickens.**

| *Escherichia coli* | | | |
|---|---|---|---|
| Number of classes in the resistant pattern | Broiler breeders (n = 1555) % | Farm broiler chickens (n = 2025) % | Slaughtered broiler chickens (n = 1178) % |
| Fully susceptible | 847 (54.5%) | 738 (36.4%) | 413 (35.1%) |
| Resistant to 1 class | 354 (22.8%) | 454 (22.4%) | 254 (21.6%) |
| Resistant to 2 classes | 237 (15.2%) | 385 (19.0%) | 254 (21.6%) |
| Resistant to 3 classes | 87 (5.6%) | 277 (13.7%) | 190 (16.1%) |
| Resistant to 4 classes | 27 (1.7%) | 137 (6.8%) | 52 (4.4%) |
| Resistant to 5 or more classes | 3 (0.2%) | 34 (1.7%) | 15 (1.3%) |
| **Multidrug-resistant isolates** | **117 (7.5%)** | **448 (22.1%)** | **257 (21.8%)** |
| *Salmonella* | | | |
| Number of classes in the resistant pattern | Broiler breeders (n = 168) % | Farm broiler chickens (n = 977) % | Slaughtered broiler chickens (n = 489) % |
| Fully susceptible | 27 (16.1%) | 412 (42.2%) | 204 (41.7%) |
| Resistant to 1 class | 119 (70.8%) | 435 (44.5%) | 232 (47.4%) |
| Resistant to 2 classes | 17 (10.1%) | 94 (9.6%) | 41 (8.4%) |
| Resistant to 3 classes | 1 (0.6%) | 35 (3.6%) | 5 (1.0%) |
| Resistant to 4 classes | 4 (2.4%) | 1 (0.1%) | 5 (1.0%) |
| Resistant to 5 or more classes | 0 (0.0%) | 0 (0.0%) | 2 (0.4%) |
| **Multidrug-resistant isolates** | **5 (3.0%)** | **36 (3.7%)** | **12 (2.4%)** |
| *Campylobacter* | | | |
| Number of classes in the resistant pattern | Broiler breeders (n = 1168) % | Farm broiler chickens (n = 465) % | Slaughtered broiler chickens (n = 677) % |
| Fully susceptible | 616 (52.7%) | 284 (61.1%) | 307 (45.4%) |
| Resistant to 1 class | 442 (37.8%) | 106 (22.8%) | 251 (37.1%) |
| Resistant to 2 classes | 102 (8.7%) | 74 (15.9%) | 112 (16.5%) |
| Resistant to 3 classes | 8 (0.7%) | 1 (0.2%) | 6 (0.9%) |
| Resistant to 4 classes | 0 (0.0%) | 0 (0.0%) | 1 (0.15%) |
| Resistant to 5 or more classes | 0 (0.0%) | 0 (0.0%) | 0 (0.0%) |
| **Multidrug-resistant isolates** | **8 (0.7%)** | **1 (0.2%)** | **7 (1.1%)** |

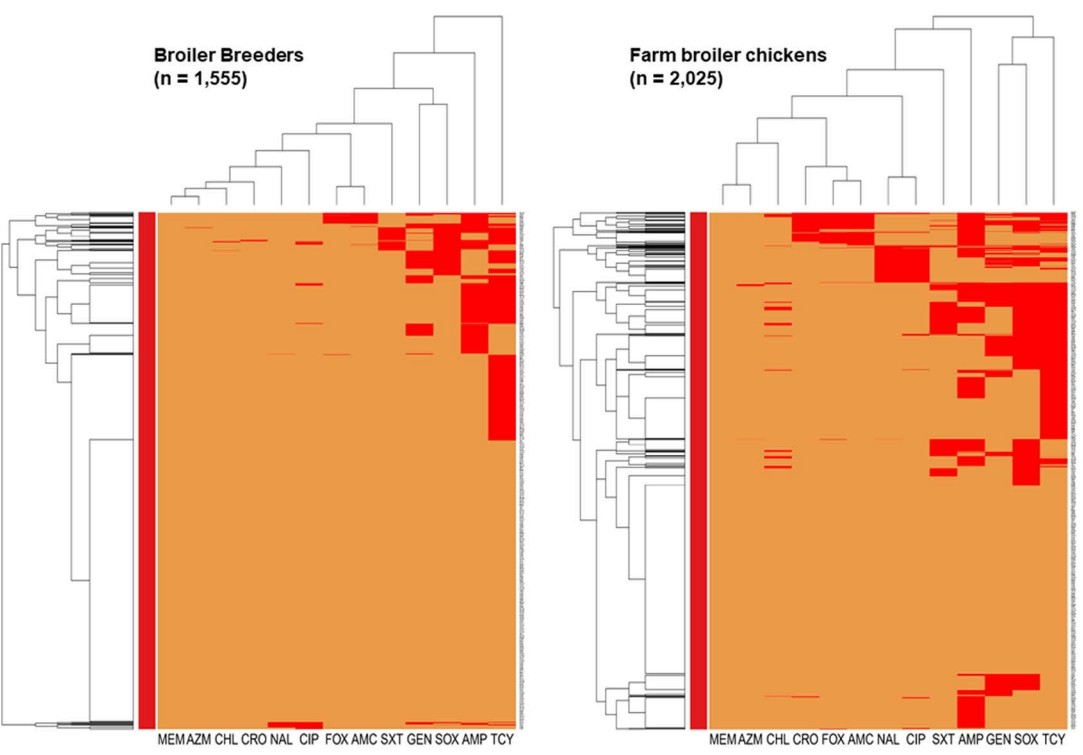

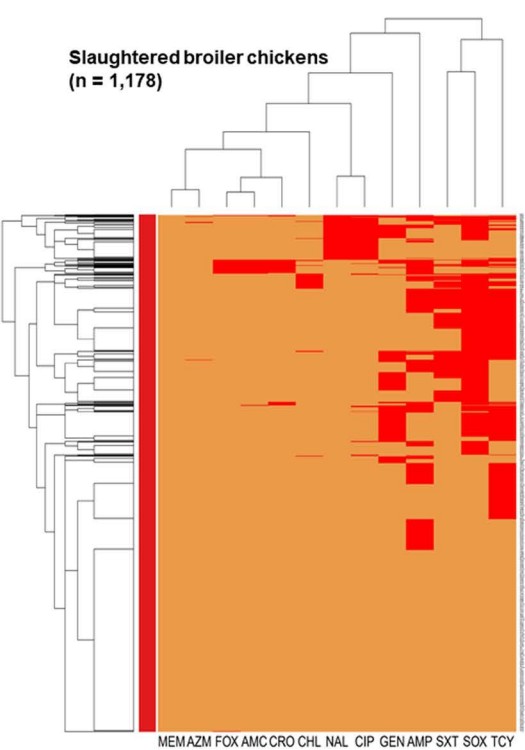

**Fig 2. Heatmap illustrating hierarchical clustering of resistance to 13 antimicrobials in generic *E. coli* (n = 1,555) from Canadian broiler breeders, farm broiler chickens (n = 2,025), and slaughtered broiler chickens (n = 1,178).** The x-axes represent antimicrobial agents: amoxicillin-clavulanic acid (AMC), ampicillin (AMP), azithromycin (AZM), chloramphenicol (CHL), ciprofloxacin (CIP)*, ceftriaxone (CRO), cefoxitin (FOX), gentamicin (GEN), meropenem (MEM), nalidixic acid (NAL), sulfisoxazole (SOX), trimethoprim-sulfamethoxazole (SXT), tetracycline (TCY). Colistin and meropenem were

omitted as there were no *E. coli* isolates resistant to colistin or meropenem. The y-axes represent the *E. coli* isolates included in this analysis. The orange colour represents susceptibility, and the red colour represents resistance. *E. coli isolates* with similar resistant patterns (based on orange-coloured cells) have shorter dendrograms than isolates that do not share as many or any similarities in their resistance profiles.

resistances were observed in the majority of E. coli isolates from Ontario and Québec. In the broiler breeder isolates, there was a small clustering of isolates resistant to sulfisoxazole and gentamicin, majority of which were received from Ontario. Further, small clusters of isolates resistant to amoxicillin-clavulanic acid and cefoxitin, found mostly in Ontario, and cipro-floxacin (non-susceptible) and nalidixic acid, found mostly in the Western region (comprised of the provinces of British Columbia, Alberta, and Saskatchewan) were observed. In farm broiler chickens, a distinct clustering among isolates resistant to tetracycline, sulfisoxazole, gentamicin and ampicillin was observed, along with clustering of resistance to sulfisoxazole and gentamicin. Additionally, clustering among isolates non-susceptible to ciprofloxacin and resistant to nalidixic acid, and isolates resistant to amoxicillin-clavulanic acid, cefoxitin, and ceftriaxone were found. The majority of the isolates from the resistant clusters in farm broiler chickens were from the Western region. In slaughtered broiler chicken isolates, there was a distinct cluster of isolates from the Western region, Ontario, and Québec resistant to sulfisox-azole, trimethoprim-sulfamethoxazole, and ampicillin. Another distinct clustering of resis-tance to gentamicin, ciprofloxacin (non-susceptible), and nalidixic acid was observed, where vast majority of the isolates were received from the Western region. Small clusters of isolates resistant to trimethoprim-sulfamethoxazole, ampicillin, and gentamicin were found mostly in Ontario and Québec. A small cluster of isolates were resistant to ceftriaxone, amoxicillin-clavulanic acid, and cefoxitin, found mostly in the Western region. This analysis shows similar resistance profiles across the three production stages but more complex profiles were observed in farm and slaughter plant isolates.

## Assessment of minimum inhibitory concentration values in *Escherichia coli*

Table 3 summarizes the MIC descriptive statistics, $MIC_{50}$ and $MIC_{90}$, and highest point across the production phases are highlighted in grey. For most antimicrobials in the panel, the $MIC_{50}$ and $MIC_{90}$ were similar across production stages, except for ampicillin and ciprofloxacin, where a lower $MIC_{50}$ (by one dilution) was observed in broiler breeders compared to farm and slaughtered broiler chickens. Additionally, higher $MIC_{90}$ were also noted from the following antimicrobials collected from farm and slaughtered broiler chickens: gentamicin (by three dilutions), and trimethoprim-sulfamethoxazole (by six dilutions).

## Salmonella

**Serovar.** More diverse *Salmonella* serovars were detected from farm broiler chickens (n = 46 unique serovars) and slaughtered chickens (n = 41) compared to broiler breeders (n = 14). The relative distribution of *Salmonella* serovars in the three production stages is shown in Fig 3 (a to c) and less frequently occurring serovars (detected in ≤ 1% of the total isolates) were aggregated. In broiler breeders, the most frequently occurring serovar was *S.* Kentucky, and distal 2nd to 4th most frequently detected were *S.* I:8,20:I:-, *S.* Heidelberg, *S.* Typhimurium var. Copenhagen and *S.* I: 8,20:-:Z6. In farm broiler chickens, the five most frequently isolated were *S.* Kentucky, *S.* Enteritidis, *S.* Liverpool, *S.* Typhimurium and *S.* Heidelberg. In slaughtered broiler chicken, the highest ranking serovars differed and included *S.* Kentucky, *S.* Enteritidis, *S.* Heidelberg, *S.* Infantis and *S.* I:8,20:-:z6. Taken together, *S.* Kentucky which oftentimes exhibits resistance to antimicrobials was the most common serovar detected in all

**Table 3. Summary of MIC$_{50}$, MIC$_{90}$ and resistance percentages of *Escherichia coli* and *Salmonella* isolates across the production stages.**

| | Antimicrobial | Production stage | *Escherichia coli* | | | | | *Salmonella* | | | | |
|---|---|---|---|---|---|---|---|---|---|---|---|---|
| | | | N | MIC$_{50}$ | MIC$_{90}$ | % R | 95% CI | n | MIC$_{50}$ | MIC$_{90}$ | % R | 95% CI |
| I | Amoxicillin-clavulanic acid | Broiler breeders | 1,555 | 4 | 8 | 2.4 | 1.7–3.3 | 168 | 1 | 32 | 11.3 | 6.9–17.1 |
| | | Farm broiler chickens | 2,025 | 4 | 8 | 6.1 | 5.1–7.2 | 977 | 1 | 2 | 9 | 7.3–11.0 |
| | | Slaughtered broiler chickens | 1,178 | 4 | 8 | 3.1 | 2.2–4.3 | 489 | 1 | 2 | 5.3 | 3.5–7.7 |
| | Colistin* | Broiler breeders | 500 | 0.25 | 0.25 | 0 | 0 | 58 | 0.25 | 0.5 | 0.6 | 0.02–3.2 |
| | | Farm broiler chickens | 907 | 0.25 | 0.25 | 0 | 0 | 381 | 0.25 | 0.5 | 0 | 0 |
| | | Slaughtered broiler chickens | 735 | 0.25 | 0.25 | 0 | 0 | 209 | 0.25 | 0.5 | 0.6 | 0.1–1.8 |
| | Ceftriaxone | Broiler breeders | 1,555 | 0.25 | 0.25 | 0.3 | 0.1–0.7 | 168 | 0.25 | 16 | 11.3 | 6.9–17.1 |
| | | Farm broiler chickens | 2,025 | 0.25 | 0.25 | 5.7 | 4.8–6.8 | 977 | 0.25 | 0.25 | 9 | 7.3–11.0 |
| | | Slaughtered broiler chickens | 1,178 | 0.25 | 0.25 | 3.6 | 2.6–4.8 | 489 | 0.25 | 0.25 | 5.7 | 3.8–8.2 |
| | Ciprofloxacin, non-susceptible | Broiler breeders | 1,555 | 0.015 | 0.015 | 2.6 | 1.8–3.5 | 168 | 0.015 | 0.015 | 0 | 0 |
| | | Farm broiler chickens | 2,025 | 0.015 | 0.03 | 8.3 | 7.1–9.5 | 977 | 0.015 | 0.03 | 4.1 | 2.9–5.5 |
| | | Slaughtered broiler chickens | 1,178 | 0.015 | 0.03 | 9.7 | 0.3–1.3 | 489 | 0.015 | 0.03 | 3.1 | 1.7–5.0 |
| | Meropenem | Broiler breeders | 1,555 | 0.06 | 0.06 | 0 | 0 | 168 | 0.06 | 0.06 | 0 | 0 |
| | | Farm broiler chickens | 2,025 | 0.06 | 0.06 | 0 | 0 | 977 | 0.06 | 0.06 | 0 | 0 |
| | | Slaughtered broiler chickens | 1,178 | 0.06 | 0.06 | 0 | 0 | 489 | 0.06 | 0.06 | 0 | 0 |
| II | Ampicillin | Broiler breeders | 1,555 | 2 | 64 | 20.3 | 18.3–22.4 | 168 | 1 | 64 | 14.9 | 9.8–21.1 |
| | | Farm broiler chickens | 2,025 | 4 | 64 | 32.1 | 30.1–34.2 | 977 | 1 | 4 | 9.8 | 8.0–11.9 |
| | | Slaughtered broiler chickens | 1,178 | 4 | 64 | 27.8 | 25.3–30.5 | 489 | 1 | 2 | 8 | 5.7–10.7 |
| | Azithromycin | Broiler breeders | 1,555 | 4 | 8 | 0.06 | 0.002–0.35 | 168 | 4 | 4 | 0 | 0 |
| | | Farm broiler chickens | 2,025 | 4 | 8 | 0.4 | 0.2–0.8 | 977 | 4 | 8 | 0 | 0 |
| | | Slaughtered broiler chickens | 1,178 | 4 | 8 | 0.3 | 0.1–0.9 | 489 | 4 | 8 | 0 | 0 |
| | Cefoxitin | Broiler breeders | 1,555 | 4 | 8 | 2.2 | 1.5–3.0 | 168 | 2 | 32 | 11.3 | 6.0–17.1 |
| | | Farm broiler chickens | 2,025 | 4 | 8 | 5.8 | 4.8–6.9 | 977 | 2 | 8 | 8.1 | 6.5–10.0 |
| | | Slaughtered broiler chickens | 1,178 | 4 | 8 | 3.1 | 2.1–4.2 | 489 | 2 | 4 | 4.9 | 3.2–7.2 |
| | Gentamicin | Broiler breeders | 1,555 | 0.5 | 4 | 9.8 | 8.3–11.4 | 168 | 0.5 | 0.5 | 0.6 | 0.02–3.2 |
| | | Farm broiler chickens | 2,025 | 0.5 | 32 | 19.6 | 17.8–21.4 | 977 | 0.25 | 0.5 | 1.5 | 0.9–2.5 |
| | | Slaughtered broiler chickens | 1,178 | 0.5 | 32 | 18.4 | 16.2–20.8 | 489 | 0.25 | 0.5 | 1.8 | 0.8–3.5 |
| | Nalidixic Acid | Broiler breeders | 1,555 | 2 | 4 | 1.1 | 0.3–1.7 | 168 | 2 | 4 | 0 | 0 |
| | | Farm broiler chickens | 2,025 | 2 | 4 | 7.7 | 6.6–9.0 | 977 | 4 | 4 | 2.7 | 1.7–3.9 |
| | | Slaughtered broiler chickens | 1,178 | 2 | 4 | 8.7 | 7.2–10.5 | 489 | 4 | 4 | 2.5 | 1.3–4.2 |
| | Trimethoprim-sulfamethoxazole | Broiler breeders | 1,555 | 0.12 | 0.12 | 4.2 | 3.3–5.4 | 168 | 0.12 | 0.12 | 1.8 | 0.3–5.1 |
| | | Farm broiler chickens | 2,025 | 0.12 | 8 | 13.4 | 11.9–14.9 | 977 | 0.12 | 0.12 | 1 | 0.5–1.9 |
| | | Slaughtered broiler chickens | 1,178 | 0.12 | 8 | 18.6 | 16.4–20.9 | 489 | 0.12 | 0.12 | 1.4 | 0.6–3.0 |
| III | Chloramphenicol | Broiler breeders | 1,555 | 4 | 8 | 0.19 | 0.04–0.6 | 168 | 4 | 8 | 2.4 | 0.65–6.0 |
| | | Farm broiler chickens | 2,025 | 8 | 8 | 3.8 | 3.0–4.7 | 977 | 4 | 8 | 0.1 | 0.002–0.5 |
| | | Slaughtered broiler chickens | 1,178 | 8 | 8 | 3.6 | 2.6–4.8 | 489 | 8 | 8 | 1.0 | 0.3–2.4 |
| | Sulfisoxazole | Broiler breeders | 1,555 | 16 | 512 | 10.7 | 9.0–12.4 | 168 | 16 | 32 | 6.6 | 3.3–11.4 |
| | | Farm broiler chickens | 2,025 | 16 | 512 | 35.1 | 33.0–37.2 | 977 | 16 | 64 | 7 | 5.4–8.7 |
| | | Slaughtered broiler chickens | 1,178 | 16 | 512 | 40.1 | 37.3–42.9 | 489 | 16 | 64 | 7.4 | 5.2–10.0 |
| | Tetracycline | Broiler breeders | 1,555 | 4 | 64 | 34.2 | 31.8–36.6 | 168 | 64 | 64 | 77.4 | 70.3–83.5 |
| | | Farm broiler chickens | 2,025 | 4 | 64 | 37.5 | 35.4–39.6 | 977 | 32 | 64 | 52.4 | 49.2–55.6 |
| | | Slaughtered broiler chickens | 1,178 | 4 | 64 | 37.1 | 34.3–40.0 | 489 | 64 | 64 | 51.9 | 47.4–56.4 |

*(Continued)*

MIC$_{50}$ (median), minimum dilution where at least 50% of isolates were inhibited.

MIC$_{90}$ (90$^{th}$ percentile) minimum dilution where at least 90% of the isolates were inhibited.

%R – percentage of resistant isolates that were inhibited based on the Clinical Laboratory Standards Institute clinical breakpoints (CLSI M100, 33$^{rd}$ Ed.) [27], where available, or based on the CIPARS clinical and US NARMS breakpoints.

Roman numeral I to III – categorization of antimicrobials according to their importance to human medicine [7].

*The addition of colistin to the panel in 2020 is reflected in the smaller number of isolates for this antimicrobial.

Grey-shaded cells – highest point/s in MIC's across the production stages for the specific antimicrobial.

production stages, with *S.* Kentucky being identified in 121 (72.0%) of broiler breeders, 458 (47.0%) of farm broiler chickens, and 182 (37.2%) of slaughtered broiler chicken isolates. Of the serovars more frequently implicated in human infections, *S.* Enteritidis ranked second most frequent serovar in farm broiler chickens and slaughtered broiler chickens, detected in 188 (19.3%) and 125 (25.6%) isolates respectively. As well, *S.* Heidelberg was detected at lower frequencies with only 7 (4.2%) broiler breeders isolates, 23 (2.4%) farm broiler chicken isolates, and 26 (5.3%) slaughtered broiler chickens isolates. *S.* Typhimurium var. Copenhagen and *S.* Typhimurium were detected in 6 (3.%) broiler breeder isolates and 30 (3.1%) farm broiler chicken isolates, respectively (Fig 3).

### Resistance percentage

For VDD Category I antimicrobials, no meropenem resistance was detected across all production stages (Fig 4). Low levels of ciprofloxacin non-susceptible isolates were detected from farm (4.1%, 95% CI: 2.9–5.5%) and slaughtered broiler chickens (3.1%, 95% CI: 1.7–5.0%) but no ciprofloxacin non-susceptible isolates were identified in broiler breeders. One colistin-resistant isolate from broiler breeders (*S.* Kentucky) and three isolates from slaughtered broiler chickens (one *S.* Kiambu and two *S.* Enteritidis) were observed. Low to moderate levels of resistance to ceftriaxone were detected across all production phases (5.7–11.3%, 95% CI: 3.5–17.1%). For the remaining antimicrobials, low-level amoxicillin-clavulanic acid (5.3–11.3%, 95% CI: 3.5–17.1%) and high to very high resistance level resistance to tetracycline (51.9–77.4%, 95% CI: 47.4–83.5%) were observed. Resistance to tetracycline was significantly lower in isolates from farm and slaughtered broiler chickens (*p* < 0.001) than from broiler breeders. Overall, unlike in *E. coli*, broiler breeders had the highest magnitude of resistance observed for most of the antimicrobials tested with higher levels detected for ampicillin and tetracycline.

### Multidrug resistance, AMR patterns, heatmap resistance distribution

Less than 50% of the isolates across all stages were fully susceptible (broiler breeders: 16.1%, 95% CI: 10.9–22.5%; farm broiler chickens: 42.2%, 95% CI: 39.0–45.3%; slaughtered broiler chickens: 41.7%, 95% CI: 37.3–46.2%). In broiler breeders, 3.0% (95% CI: 0.9–6.8%) of isolates were MDR, while 3.7% (95% CI: 2.6–5.1%) of farm broiler chicken and 2.4% (95% CI: 1.3–4.2%) of slaughtered broiler chicken isolates were MDR (Table 2).

Resistance to tetracycline was the most common resistance pattern in broiler breeders (65.5%, 95% CI: 57.8–72.6%) and slaughtered broiler chickens (42.1%, 95% CI: 37.7–46.6%), while this was the second most common pattern in farm broiler chickens (40.3%, 95% CI: 37.2–43.5%). Fully susceptibility was the most common profile of farm broiler chickens (42.2%, 95% CI: 39.0–45.3%), while this was the second most common pattern in broiler breeders (16.1%, 95% CI: 10.9–22.5%) and slaughtered broiler chickens (41.7%, 95% CI: 37.3–46.2%). The third most common pattern in broiler breeders and farm broiler chickens

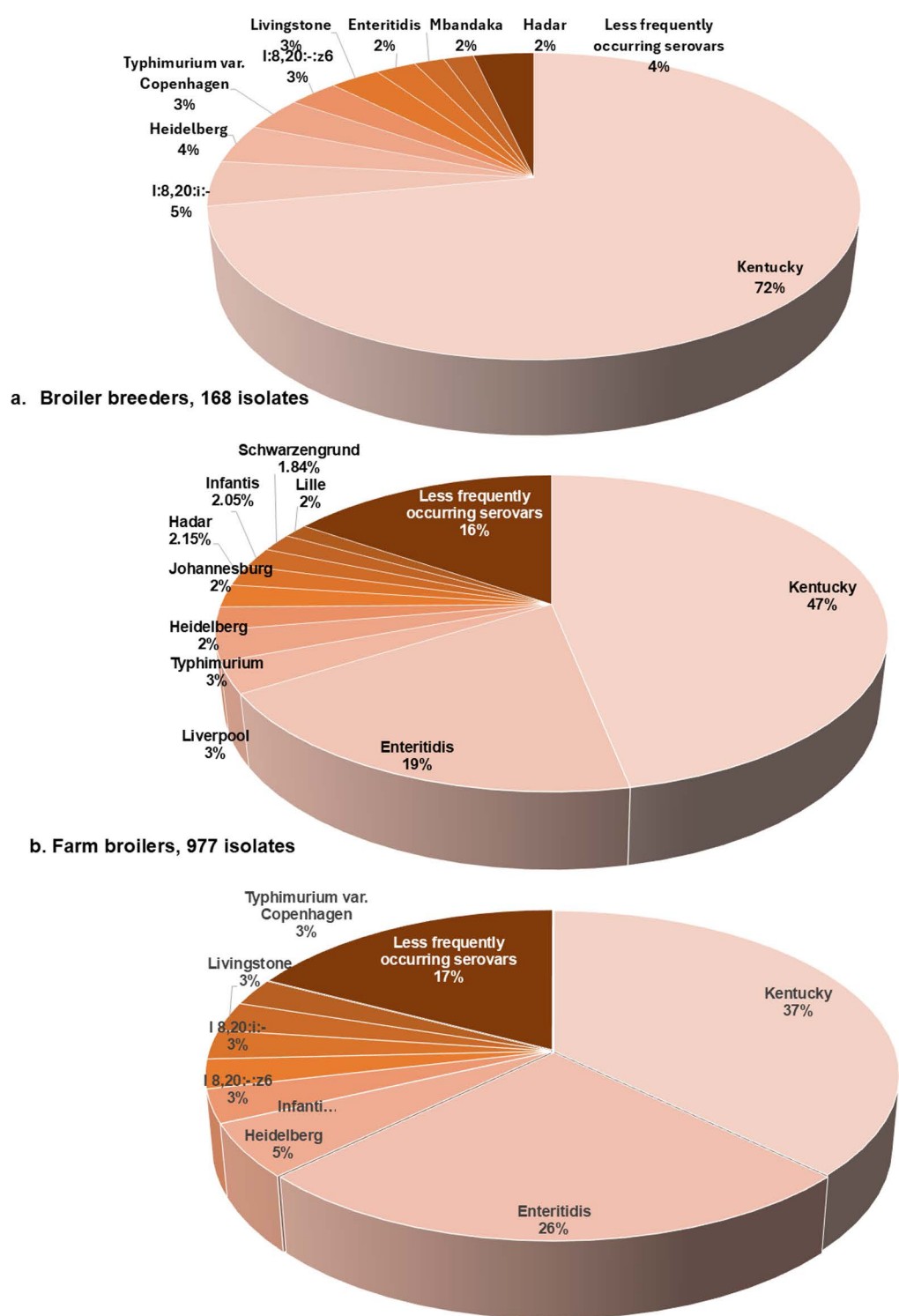

a. Broiler breeders, 168 isolates

b. Farm broilers, 977 isolates

c. Slaughtered broilers, 489 isolates

**Fig 3. Number of *Salmonella* serovars recovered from broiler breeders (n = 168), farm broiler chickens (n = 977) and slaughtered broiler chickens (n = 489), 2018–2021.** Less-frequently occurring serovars detected at ≤ 1% of the total isolates were combined including six other serovars from broiler breeders, 36 other serovars from farm broiler chickens and 33 other serovars from slaughtered broiler chickens.

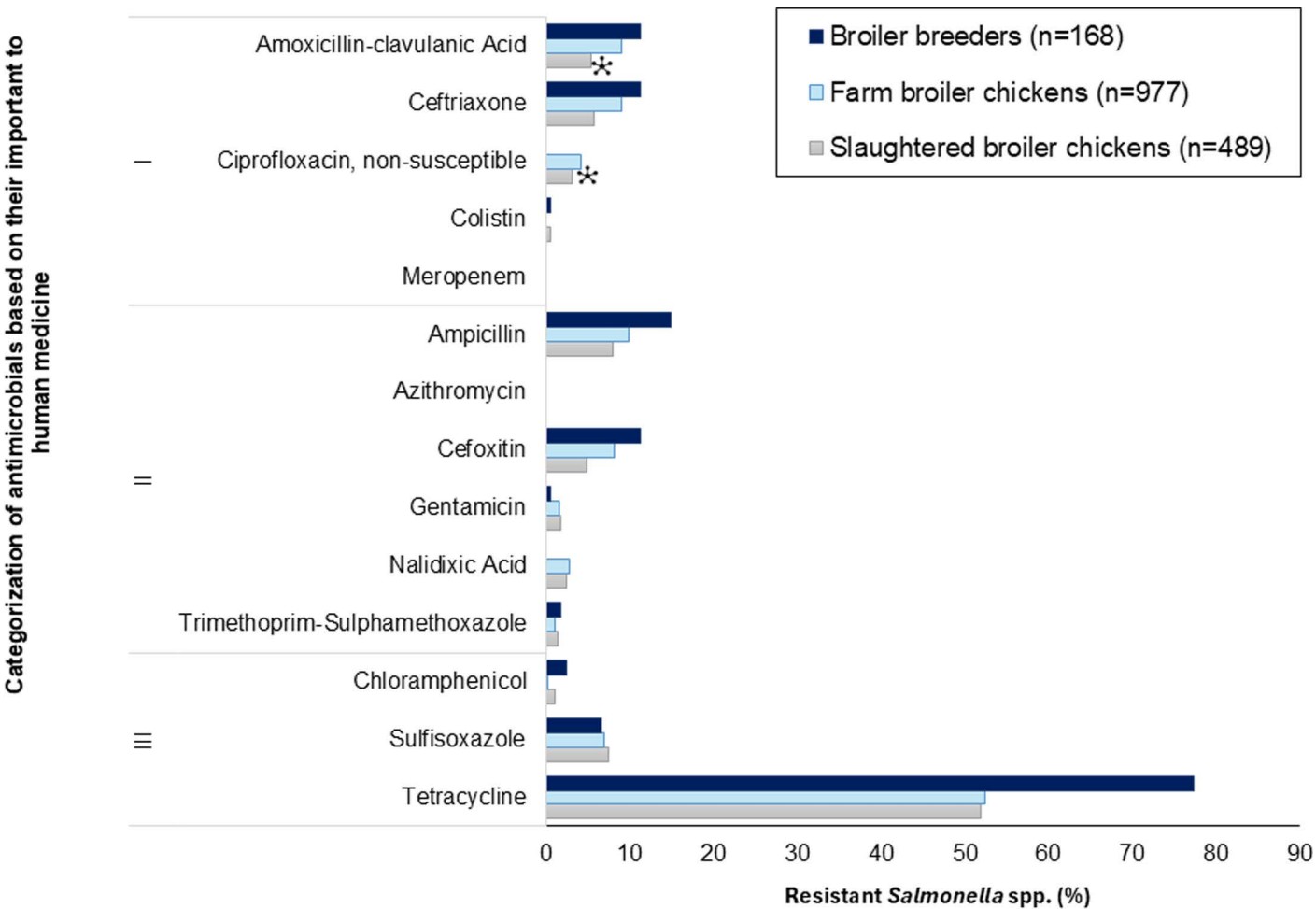

**Fig 4. Percentage of *Salmonella* isolates resistant to tested antimicrobials from broiler breeders, farm broiler chickens, and slaughtered broiler chickens.**
\* significantly lower vs. broiler breeders (the referent) ($p \leq 0.05$).

was amoxicillin-clavulanic acid-ampicillin-cefoxitin-ceftriaxone-tetracycline at 7.1% (95% CI: 3.7–12.1%) and 4.0% (95% CI: 2.9–5.1%), respectively. Sulfisoxazole-tetracycline was the third most common pattern in slaughtered broiler chickens at 4.5% (95% CI: 2.8–6.7%)

To detect more specific clustering of AMR by the most common serovar within regions, the resistance distribution of *S.* Kentucky, the most predominant isolate across the production continuum and across geographical regions was mapped. Across all three production phases, in the Western provinces and in Québec, there was a distinct clustering of *S.* Kentucky isolates resistant to tetracycline and ampicillin. While there was a small group of isolates also resistant to ampicillin, amoxicillin-clavulanic acid, ceftriaxone, and cefoxitin in all phases and regions except for farm broiler chickens in Ontario. Clustering of resistant isolates to ampicillin, amoxicillin-clavulanic acid, ceftriaxone, cefoxitin, ciprofloxacin, and nalidixic acid was also observed in farm and slaughtered broiler chickens in the Western region (Fig 5).

## Assessment of minimum inhibitory concentration values in *Salmonella*

Table 3 summarizes the MIC descriptive statistics for *Salmonella*. Grey shaded cells are highest point across the production phases. For most antimicrobials in the panel, the MIC$_{50}$

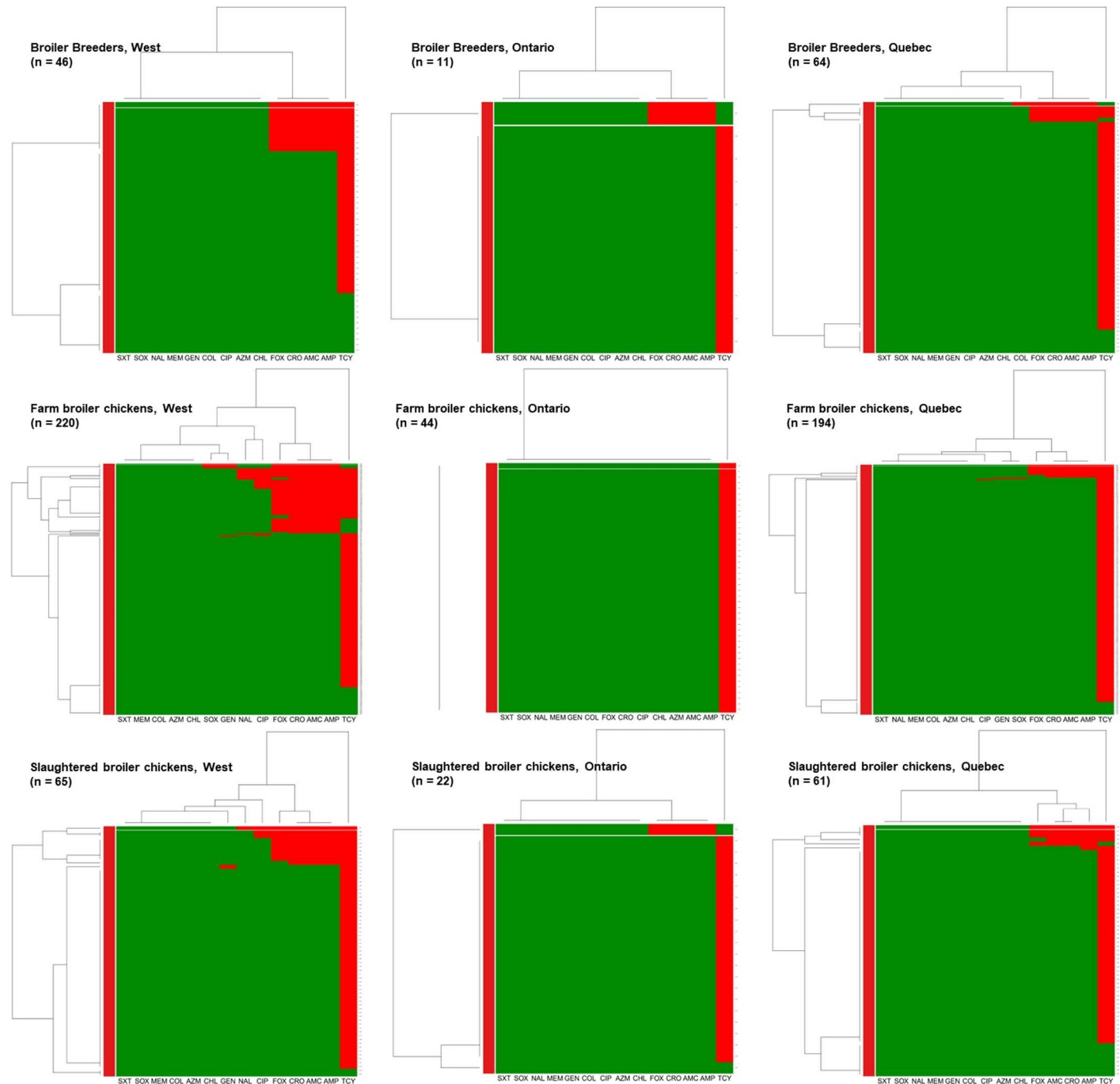

**Fig 5. Heatmap illustrating hierarchical clustering of resistance to 14 antimicrobials in *S.* Kentucky from Canadian broiler breeders in the West (n = 46), Ontario (n = 11) and Québec (n = 64) regions, farm broiler chicken in the West (n = 220), Ontario (n = 44) and Québec (n = 194) regions, and slaughtered broiler chickens in the West (n = 65), Ontario (n = 22) and Québec (n = 61) regions.** X-axes represent antimicrobials: amoxicillin-clavulanic acid (AMC), ampicillin (AMP), azithromycin (AZM), chloramphenicol (CHL), ciprofloxacin (CIP)\*, ceftriaxone (CRO), cefoxitin (FOX), colistin (COL), gentamicin (GEN), meropenem (MEM), nalidixic acid (NAL), sulfisoxazole (SOX), trimethoprim-sulfamethoxazole (STX), tetracycline (TCY). Y-axes represent the *S.* Kentucky isolates included in this analysis. The green colour represents susceptibility, and the red colour represents resistance. *Salmonella* isolates with similar resistance patterns (based on red-coloured cells) have shorter dendrograms than isolates that do not share as many or any similarities in their resistance profiles.

and $MIC_{90}$ were higher in broiler breeders such as gentamicin, where broiler breeder $MIC_{50}$ was higher by one dilution compared to farm and slaughtered broiler chickens. Also, higher $MIC_{90}$ was observed for ampicillin in broiler breeders compared to farm (by four dilutions) and slaughtered broiler chickens (by five dilutions). Similar patterns were noted for ceftriaxone (by six dilutions), ampicillin (by three to four dilutions) and cefoxitin (by two to three dilutions). It is important to note that the $MIC_{50}$ and $MIC_{90}$ for ciprofloxacin were the same (0.015 μg/ml) in broiler breeders, but one dilution higher for the $MIC_{90}$ values for farm and slaughtered broiler chickens (Table 3).

## Campylobacter

**Resistance percentage.** High levels of resistance to ciprofloxacin were observed across all production stages, where 24.0% (95% CI: 21.5–26.5%) of broiler breeder isolates, 20.0% (95% CI: 16.5–24.0%) of farm broiler chicken isolates, and 21.6% (95% CI: 18.5–24.9%) of slaughtered broiler chicken isolates were resistant. Of note, 87.9% (95% CI: 25.5–32.5%) of *Campylobacter* spp. isolates from broiler breeders, and 100% (95% CI: 47.8–100%) of slaughtered broiler chicken isolates were resistant to ciprofloxacin (Table 4). Moderate to high levels of resistance to nalidixic acid, observed across the stages, mirrored the ciprofloxacin resistance levels (broiler breeders: 24.0%, 95% CI: 21.5–26.5%; farm broiler chickens: 20.0%, 95% CI: 16.5–23.9; slaughtered broiler chickens: 21.6%, 95% CI: 18.5–24.9%). Nalidixic acid-resistant *Campylobacter* spp. isolates also mirrored ciprofloxacin resistance levels. High levels of tetracycline resistance were also detected across all stages. Slaughtered broiler chickens had a significantly higher percentage of tetracycline-resistant isolates compared to broiler breeders (44.9%, 95% CI: 41.1–48.7% and 30.7%, 95% CI: 28.1–33.5 respectively, *p* < 0.001). While 30.8% (95% CI: 26.6–35.2%) of farm broiler chickens were resistant to tetracycline (Fig 6).

## Multidrug resistance, AMR patterns, heatmap resistance distribution

All stages were observed to have fully susceptible *Campylobacter* isolates, where 52.7% (95% CI: 49.8–55.6%), 61.1% (95% CI: 56.5–65.5%), and 45.4% (95% CI: 41.6–49.2%) of broiler breeder, farm, and slaughtered broiler chicken isolates, respectively were fully susceptible. Multidrug-resistant *Campylobacter* were observed at lower frequencies compared to the Enterobacterales (*E. coli* and *Salmonella* spp.) across all stages [broiler breeders: 0.7% (95% CI: 0.2–1.4%); farm broiler chickens: 0.2% (95% CI: 0.01–1.2%); slaughtered broiler chickens: 1.1% (95% CI: 0.4–2.1%)] (Table 2).

As described above, the most common resistance pattern observed in *Campylobacter* isolates was full susceptibility to the eight antimicrobials which was noted across all production stages. Other patterns include tetracycline resistance (broiler breeders: 22.2% (95% CI: 19.8–24.7%); farm broiler chickens: 16.6% (95% CI: 13.3–20.3%); slaughtered broiler chickens: 29.4% (95% CI: 26.0–33.0%)), ciprofloxacin-nalidixic acid resistance in broiler breeder isolates (15.6%, 95% CI: 13.5–17.8%) and ciprofloxacin-nalidixic acid-tetracycline resistance in farm, and slaughtered broiler chicken isolates (14.0%, 95% CI: 11.0–17.5% and 13.9%, 95% CI: 11.4–16.7% respectively).

Among the broiler breeder isolates, a distinct cluster of isolates resistant to nalidixic acid and ciprofloxacin, and another cluster resistant to tetracycline, nalidixic acid, and ciprofloxacin were present. The majority of the isolates from both clusters were from the Western region. A small cluster of isolates were resistant to erythromycin, azithromycin, and clindamycin, the majority of which were found in Ontario. Isolates from the farm broiler chickens and slaughter broiler chickens showed distinct clustering of resistance to tetracycline, nalidixic acid, and ciprofloxacin, where the majority of the isolates were found in the Western region.

**Table 4. Summary of MIC$_{50}$, MIC$_{90}$ and resistance percentages of *Campylobacter* isolates across the production stages.**

*Campylobacter*

| | Antimicrobial | Production Stage | Species | n | MIC$_{50}$ | MIC$_{90}$ | % R | 95% CI |
|---|---|---|---|---|---|---|---|---|
| I | Ciprofloxacin | Broiler Breeders | *C. jejuni* | 470 | 0.12 | 8 | 12.6 | 9.7–15.9 |
| | | | *C. coli* | 665 | 0.12 | 16 | 28.9 | 25.5–32.5 |
| | | | *C. spp* | 33 | 4 | 8 | 87.9 | 71.8–96.6 |
| | | Farm broiler chickens | *C. jejuni* | 412 | 0.12 | 16 | 17.7 | 14.2–21.8 |
| | | | *C. coli* | 53 | 0.12 | 16 | 37.7 | 24.8–52.1 |
| | | Slaughtered broiler chickens | *C. jejuni* | 581 | 0.12 | 16 | 20 | 16.8–23.5 |
| | | | *C. coli* | 91 | 0.12 | 8 | 27.5 | 18.6–37.8 |
| | | | *C. spp* | 5 | 4 | 4 | 100 | 47.8–100 |
| II | Azithromycin | Broiler Breeders | *C. jejuni* | 470 | 0.03 | 0.06 | 0.4 | 0.1–1.5 |
| | | | *C. coli* | 665 | 0.12 | 0.25 | 2.3 | 1.3–3.7 |
| | | Farm broiler chickens | *C. jejuni* | 412 | 0.06 | 0.06 | 0 | 0 |
| | | | *C. coli* | 53 | 0.06 | 128 | 20.8 | 10.8–34.1 |
| | | Slaughtered broiler chickens | *C. jejuni* | 581 | 0.03 | 0.06 | 2.9 | 1.7–4.6 |
| | | | *C. coli* | 91 | 0.06 | 128 | 11 | 5.4–19.3 |
| | Clindamycin | Broiler Breeders | *C. jejuni* | 470 | 0.12 | 0.25 | 0.2 | 0.001–1.2 |
| | | | *C. coli* | 665 | 0.25 | 0.5 | 2 | 1.0–3.3 |
| | | Farm broiler chickens | *C. jejuni* | 412 | 0.12 | 0.25 | 0 | 0 |
| | | | *C. coli* | 53 | 0.25 | 8 | 18.9 | 9.4–32.0 |
| | | Slaughtered broiler chickens | *C. jejuni* | 581 | 0.12 | 0.25 | 1.7 | 0.8–3.1 |
| | | | *C. coli* | 91 | 0.25 | 8 | 11 | 5.4–19.3 |
| | Erythromycin | Broiler Breeders | *C. jejuni* | 470 | 0.25 | 0.5 | 0.4 | 0.1–1.5 |
| | | | *C. coli* | 665 | 0.5 | 1 | 2.3 | 1.3–3.7 |
| | | Farm broiler chickens | *C. jejuni* | 412 | 0.25 | 0.5 | 0 | 0 |
| | | | *C. coli* | 53 | 0.5 | 128 | 20.8 | 10.8–34.1 |
| | | Slaughtered broiler chickens | *C. jejuni* | 581 | 0.25 | 0.5 | 2.9 | 1.7–4.6 |
| | | | *C. coli* | 91 | 0.5 | 64 | 11 | 5.4–19.3 |
| | Gentamicin | Broiler Breeders | *C. jejuni* | 470 | 1 | 1 | 0 | 0 |
| | | | *C. coli* | 665 | 1 | 1 | 0 | 0 |
| | | Farm broiler chickens | *C. jejuni* | 412 | 1 | 1 | 0 | 0 |
| | | | *C. coli* | 53 | 0.5 | 1 | 0 | 0 |
| | | Slaughtered broiler chickens | *C. jejuni* | 581 | 1 | 1 | 0 | 0 |
| | | | *C. coli* | 91 | 0.5 | 1 | 0 | 0 |
| | Nalidixic Acid | Broiler Breeders | *C. jejuni* | 470 | 4 | 128 | 12.6 | 9.7–15.9 |
| | | | *C. coli* | 665 | 8 | 128 | 28.9 | 25.5–32.5 |
| | | | *C. spp* | 33 | 128 | 128 | 87.9 | 71.8–96.6 |
| | | Farm broiler chickens | *C. jejuni* | 412 | 4 | 128 | 17.7 | 14.2–21.8 |
| | | | *C. coli* | 53 | 8 | 128 | 37.7 | 24.8–52.1 |
| | | Slaughtered broiler chickens | *C. jejuni* | 581 | 4 | 128 | 20 | 16.8–23.5 |
| | | | *C. coli* | 91 | 8 | 128 | 27.5 | 18.6–37.8 |
| | | | *C. spp* | 5 | 128 | 128 | 100 | 47.8–100[a] |
| III | Florfenicol | Broiler Breeders | *C. jejuni* | 470 | 1 | 1 | 0 | 0 |
| | | | *C. coli* | 665 | 1 | 1 | 0 | 0 |
| | | Farm broiler chickens | *C. jejuni* | 412 | 1 | 2 | 0 | 0 |
| | | | *C. coli* | 53 | 1 | 2 | 0 | 0 |
| | | Slaughtered broiler chickens | *C. jejuni* | 581 | 1 | 1 | 0 | 0 |
| | | | *C. coli* | 91 | 1 | 2 | 0 | 0 |

*(Continued)*

**Table 4.** (Continued)

*Campylobacter*

| | Antimicrobial | Production Stage | Species | n | MIC$_{50}$ | MIC$_{90}$ | % R | 95% CI |
|---|---|---|---|---|---|---|---|---|
| | Tetracycline | Broiler Breeders | *C. jejuni* | 470 | 1 | 128 | 49.2 | 44.5–53.7 |
| | | | *C. coli* | 665 | 0.25 | 128 | 19.3 | 16.3–22.5 |
| | | Farm broiler chickens | *C. jejuni* | 412 | 0.25 | 128 | 32.8 | 28.3–37.5 |
| | | | *C. coli* | 53 | 0.5 | 64 | 15.1 | 6.7–27.6 |
| | | Slaughtered broiler chickens | *C. jejuni* | 581 | 0.5 | 128 | 46.6 | 42.5–50.8 |
| | | | *C. coli* | 91 | 0.5 | 128 | 36.3 | 26.4–47.0 |

MIC$_{50}$ [median], minimum dilution where at least 50% of isolates were inhibited.

MIC$_{90}$ [90th percentile] minimum dilution where at least 90% of the isolates were inhibited.

%R – percentage of resistant isolates that were inhibited based on the Clinical Laboratory Standards Institute clinical breakpoints (CLSI M45) [28], where available, or based on the CIPARS clinical and NARMS breakpoints.

Roman numeral I to III – categorization of antimicrobials according to their importance to human medicine [7].

A small cluster of isolates were resistant to erythromycin, azithromycin, and clindamycin, the majority of which were from Québec (Fig 7).

## Assessment of minimum inhibitory concentration values in *Campylobacter*

Table 4 summarized the MIC descriptive statistics and grey shaded cells depict the highest point across the production phases for each antimicrobial-*Campylobacter* species combination. In most cases, the highest observed MIC$_{90}$ were noted in *C. coli* compared to *C. jejuni*. Broiler breeder and farm *C. coli* isolates had higher MIC$_{90}$ in ciprofloxacin (by one dilution) compared to slaughter plant *Campylobacter* isolates. As well, farm and slaughtered chicken *C. coli* isolates had the highest MIC$_{90}$ values in azithromycin (nine dilutions higher) and clindamycin (four dilutions higher). Tetracycline MIC$_{90}$ values were similar across the production phases and *Campylobacter* species, except in farm broiler chicken isolates (one dilution lower than breeders and slaughtered chickens). It is important to note that the limited number of unspeciated *Campylobacter* (*Campylobacter* spp.) isolates from broiler breeders (n = 33) and slaughtered broiler chickens (n = 5) exhibited the highest MIC$_{50}$ for ciprofloxacin (five dilutions) and nalidixic acid (four to five dilutions) compared to *C. jejuni* and *C. coli*.

## Discussion

This work provides a descriptive overview of the recovery and AMR prevalence among *E. coli, Salmonella,* and *Campylobacter* isolates from three stages of broiler chicken production, from the broiler breeders (supplying hatching eggs) down to the broiler chickens at the farm and slaughter plant levels. The results indicate that AMR to VDD Category I antimicrobials (antimicrobials of very high importance to human medicine) were low in the Enterobacterales (*E. coli* and *Salmonella*) across the poultry production chain. However, we observed high levels of ciprofloxacin resistance in *Campylobacter* isolates.

High levels of ciprofloxacin resistance (≥ 20% of the isolates) were observed in *Campylobacter* across all three production stages indicating the widespread distribution of *Campylobacter* spp. resistant to ciprofloxacin in the poultry production continuum. This is substantially higher than what has been reported by Huber et al. [24] in farm broiler chickens (16.5%) and in slaughtered broiler chickens at the slaughter plant in Ontario (0% to 16%) during the early implementation phases of the farm and slaughter plant programmes (between 2010 and 2015) [15]. However, the high levels of ciprofloxacin resistance are comparable to resistance levels in

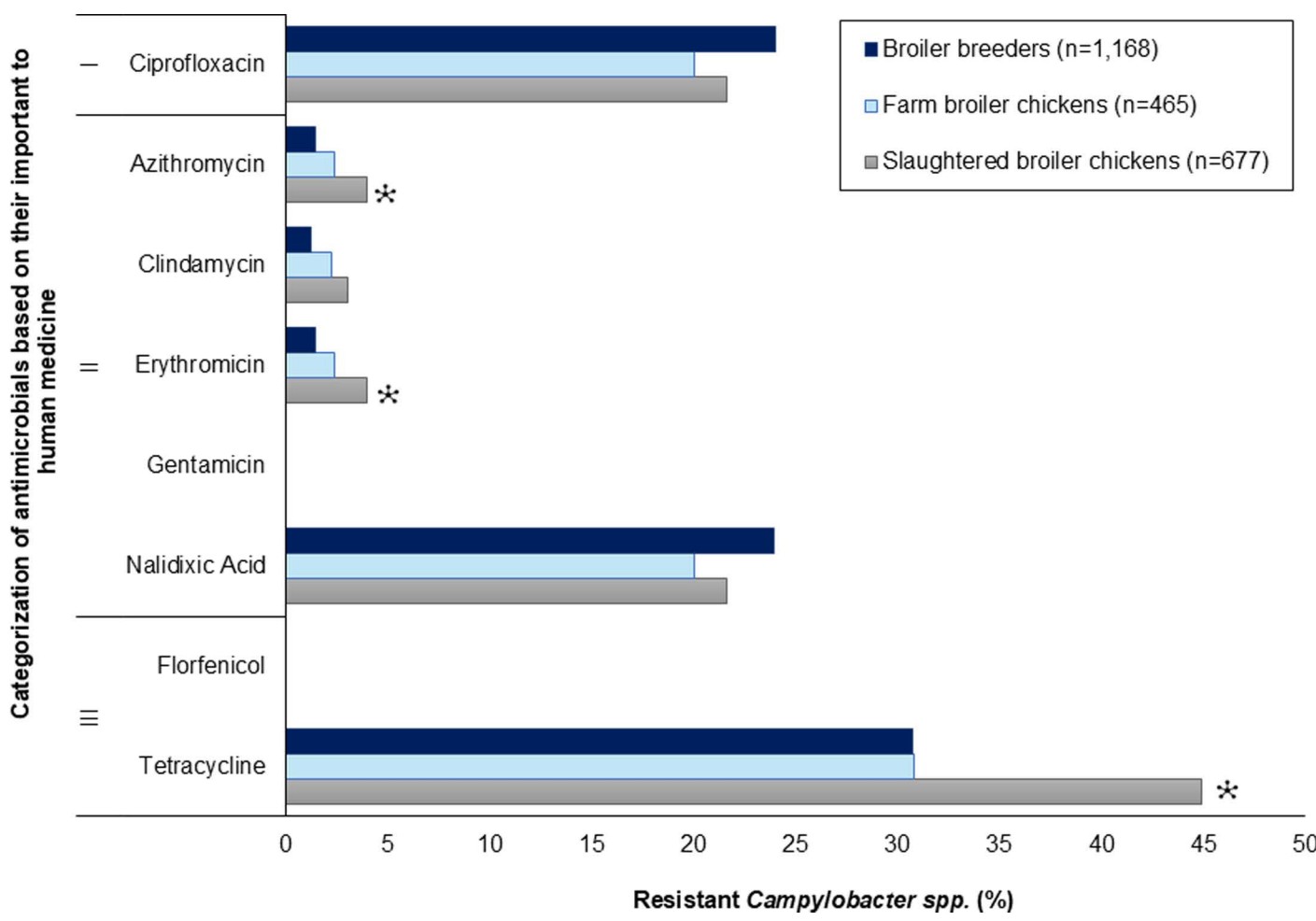

**Fig 6. Percentage of resistance in *Campylobacter* from broiler breeders, farm broiler chickens, and slaughtered broiler chickens.** * significantly higher vs. broiler breeders (the referent) ($p \leq 0.05$).

farm broiler chickens as reported by the CIPARS group [15] between 2013 to 2015 (5% to 33%), which is suggestive that *Campylobacter* resistant to ciprofloxacin has persisted in the broiler chicken production chain. CIPARS does not collect information regarding AMU in broiler breeders and no data were collected using the "flock sheets" but there is anecdotal evidence that fluoroquinolones are being used. Because potential vertical transmission of fluoroquinolone-resistant *C. coli* could occur in nature [21], and persistent colonization of these isolates could lead to its dissemination in the ecosystem, therefore even a limited number of broiler breeder flocks exposed to fluoroquinolones might influence the dissemination and maintenance of fluoroquinolone-resistant *Campylobacter* to broiler chickens. In many instances [4,21,33] ciprofloxacin-resistant isolates were also recovered from raw chicken meats at the point of sale (retail), the production phase closest to the consumer. In addition to ciprofloxacin, high level of tetracycline resistance was prevalent in *Campylobacter* isolates across all three production stages, where it was significantly higher in slaughtered broiler chickens than in broiler breeders, possibly due to recent AMU exposures or other sources of resistant *Campylobacter* at the broiler farm level, for example, the retention of *Campylobacter* resistant to antimicrobials from previous flock cycles [15]. The CIPARS Veterinary Antimicrobial Sales Reporting (VASR)

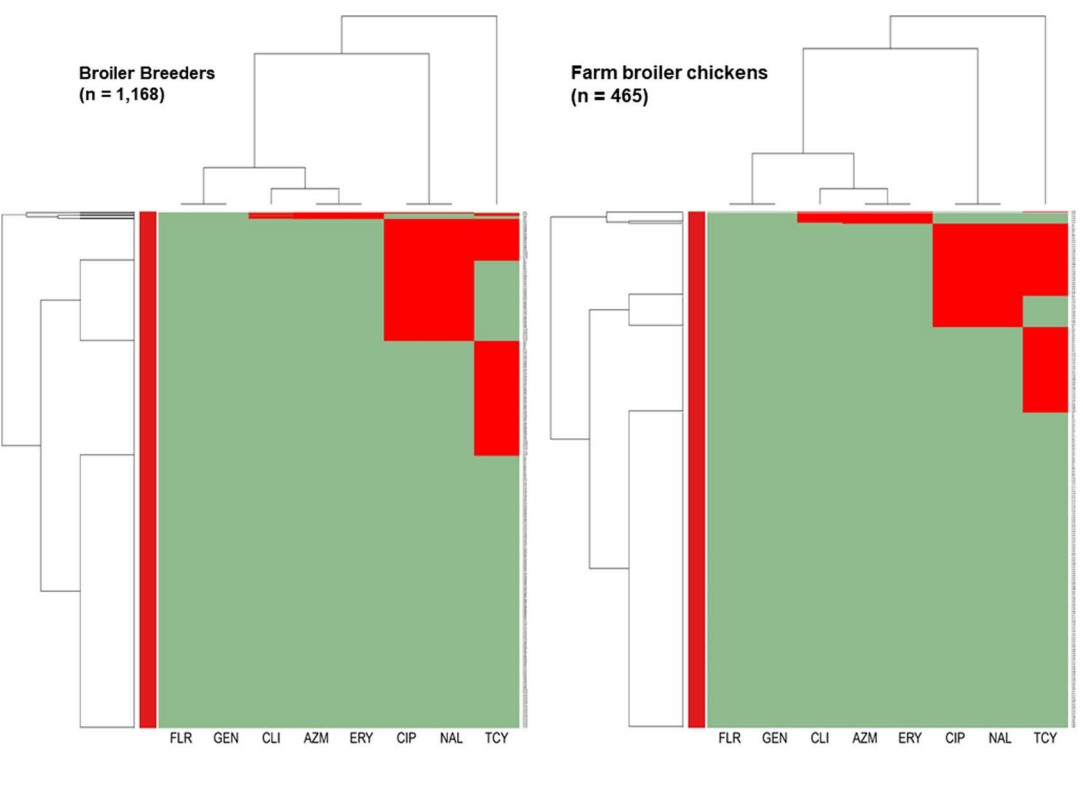

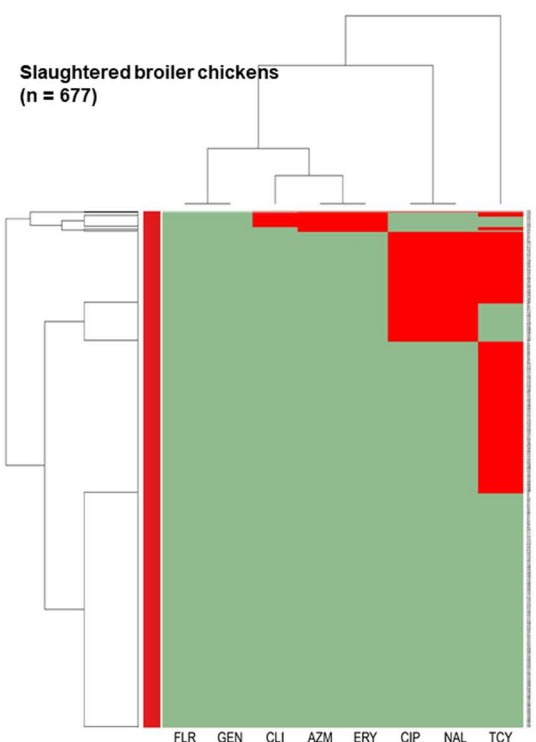

**Fig 7. Heatmap illustrating hierarchical clustering of resistance to eight antimicrobials in** *Campylobacter* **from Canadian broiler breeders (n = 1,168), farm broiler chickens (n = 465), and slaughtered broiler chickens (n = 677).** X-axes represent antimicrobial agents: azithromycin (AZM), ciprofloxacin (CIP), clindamycin (CLI), erythromycin (ERY), florfenicol (FLR), gentamicin (GEN), nalidixic acid (NAL), and tetracycline (TCY). Y-axes represent the *Campylobacter* isolates included in this analysis. The light green colour represents susceptibility, and the red colour represents resistance.

reported the continued use of tetracycline in poultry and this was also reported in the farm program during the timeframe of this study [38]. Although we observed high ciprofloxacin resistance across the poultry production continuum, there are limited studies demonstrating the vertical transfer of *Campylobacter* [39–41]. Previous studies have concluded that vertical transmission of *Campylobacter* in broiler chickens is a low-risk route of exposure [39]. The use of fluoroquinolones (including ciprofloxacin) for veterinary use is available in Canada by prescription only [42]. In chickens, they are used for rare bacterial disease outbreaks or to treat severe infections where other antimicrobials have been ineffective [43]. Although CIPARS have not documented high use of fluoroquinolones in the broiler chicken sector, occasional flocks are treated (with enrofloxacin, a veterinary fluoroquinolone) and the persistence of *Campylobacter* resistant to ciprofloxacin in the environment may be contributing to the sustained level of ciprofloxacin-resistant *Campylobacter* in broiler chickens [44]. Additionally, resistance to tetracyclines has been associated with ciprofloxacin resistance [43–45], and the association was also observed in the current study. CIPARS authors previously hypothesized [44], based on a related study [46] that a mutation in the *gyrA* gene, and exposure to ciprofloxacin and ciprofloxacin-tetracycline (or different antimicrobial active ingredients belonging to the same classes) resulting in the overexpression of the multidrug-resistant efflux pump CmeABC, contribute to the emergence of ciprofloxacin resistance in *Campylobacter*. The relationship between the detection of these AMR patterns and underlying genetic mechanisms would be important to characterize. Of further concern was the presence of unspeciated *Campylobacter* species in breeders and in broilers at slaughter showing an unusually high proportion of resistance to ciprofloxacin. Additional analysis of these isolates (beyond PCR-based speciation) would be necessary to better understand the importance of this observation.

Overall, there was a low prevalence of ceftriaxone resistant *E. coli* across the Canadian poultry production continuum (lower than 10% across the three production stages). This is a positive observation considering the ban the Canadian industry has imposed on the preventative use of cephalosporins in broiler chickens in 2014. Given the risk of vertical transfer of cephalosporin-resistant *E. coli*, similar efforts to voluntarily ban the use of ceftiofur in broiler breeders originating from the United States should be pursued.

Resistance to another Category I antimicrobial, amoxicillin-clavulanic acid and non-susceptibility to ciprofloxacin in *E. coli* isolates, were less pronounced in broiler breeders compared to farm and slaughtered broiler chickens, which may indicate that breeders may also play a role in the downstream dissemination of resistant isolates. The changes in AMR profiles downstream in the food production chain (farm and slaughter plant levels) could signify other sources of exposures such as AMU and cross-contamination/carry-over from previous flocks as well as fomites present in the farm [11]. This has been demonstrated in a previous study in Ontario where chicks at placement represent potential AMR profiles of breeders and hatchery-level AMR sources [15]. Further, we observed moderate to high levels of resistance to ampicillin and sulfisoxazole in farm and slaughtered broiler chickens which were significantly higher than in broiler breeders. Antimicrobials belonging to the same class (penicillins and sulfonamides) were used in broiler flocks during the study timeframe [45], which could explain the resistance levels observed. Noting the difference in defined breakpoints, we report the high levels of non-susceptibility to ciprofloxacin (based on intermediate CLSI breakpoint of $\geq 0.12$ μg/mL) [27] in *E. coli* isolates, compared to ciprofloxacin resistance reported in previous analysis of farm [24] and slaughtered [47] chicken isolates based on the CLSI breakpoint for resistant isolates ($\geq 1$ μg/mL) [27]. Monitoring of non-susceptible ciprofloxacin *E. coli* could be used to identify an emerging or persistent AMR issue such as fluoroquinolone-resistant Enterobacterales in Canadian poultry production. Resistance to amoxicillin-clavulanic acid, ampicillin, sulfisoxazole and tetracycline was lower in this current

analysis than what was reported in the former studies [24,47]. Specifically, we observed a more pronounced decrease in resistance than what was observed by Huber et al. [24] for the aforementioned older antimicrobials, following the AMU reduction initiatives in 2014 led by the Canadian poultry industry [8]. The Canadian poultry industry initiative called for the voluntary elimination of the preventive use of Category I antimicrobials (such as 3$^{rd}$ generation cephalosporins and fluoroquinolones) in May 2014 and Category II antimicrobials (such as aminoglycosides, aminocyclitol/lincosamide, macrolides, penicillins, and streptogramins) at the end of 2018 [8]. At the time of this current study (2018 to 2022), the industry had already implemented the elimination of the preventive use of injectable antimicrobials such as ceftiofur, and was in the early implementation phase of the preventive use of gentamicin and lincomycin-spectinomycin at the hatchery.

The absence of exposure to 3$^{rd}$ generation cephalosporins might explain sustained low-level resistances to ceftriaxone observed in our study. Limited studies have assessed the risk factors for extended-spectrum beta-lactamases (ESBLs) and/or AmpC occurrence in animals, however, there are some possible contributing factors to the spread of ESBLs and/or AmpC-producing bacterial strains in food animals [48]. Potential contributing factors include farm management factors such as the stocking of new animals potentially exposed (including via international trade of animals), exposure to contaminated air, water, feed, insect or rodent vectors, as well as human-to-animal transmission [48]. It is also suspected that if ESBLs or AmpC-producing bacteria are already present in the production phase, the use of various antimicrobials may facilitate their selection and the spread of resistant organisms between animals [48]. Resistance to antimicrobials such as amoxicillin, amoxicillin-clavulanic acid, tetracycline, trimethoprim, sulphonamides, quinolones, and phenicols have been linked to ESBLs and AmpC-producing bacteria [49–52] – which is observed in the current study. Some of these antimicrobials (amoxicillin, tetracycline, trimethoprim and sulfonamides) are reportedly used in broiler chickens in Canada [45].

Similar to *E. coli* isolates, *Salmonella* isolates were observed to have low levels of resistance to Health Canada's VDD Category I antimicrobials, although broiler breeders showed higher levels of resistance to most antimicrobials compared to farm and slaughtered broiler chickens. The presence of a colistin-resistant *S.* Kentucky in broiler breeders is a source of concern given that drug-resistant *Salmonella* are transmitted vertically [53] and that such strain might eventually find its way to the other chicken production levels. Colistin-resistant isolates in broiler breeders and slaughtered broiler chickens were rarely detected through CIPARS, and their exact source/s is unknown. Colistin-resistant *Salmonella* were also rarely detected in other animal species sampled by CIPARS; to date, no mobile colistin *Salmonella* isolates have been found from animal or food sources by CIPARS [54]. Compared to previous Canadian studies that reported no resistance to ciprofloxacin based on the CLSI clinical breakpoint, this current study, reports non-susceptible ciprofloxacin *Salmonella* isolates based on the presence of genes conferring resistance to fluoroquinolones by WGS, enabling the early detection of the emergence of fluoroquinolone-resistant Enterobacterales. The drivers for the emergence of non-susceptible ciprofloxacin *Salmonella* are unknown as there are a limited number of broiler flocks reportedly treated with fluoroquinolones during the entire CIPARS farm surveillance timeframe [4,45] and no flocks reportedly treated during the timeframe for this analysis. The differences (between susceptible and non-susceptible ciprofloxacin-resistant isolates warrants further investigation using molecular methods. Also compared with previous studies [24,47], higher levels of resistance to tetracycline were found in our study, while resistance to amoxicillin-clavulanic acid, and ceftriaxone were found to be similar or lower. The diversity of *Salmonella* serovars varied depending on the production stage, but *S.* Kentucky was the most common serovar found across all three production

stages. Although this serovar has not been reported to be a serovar that commonly causes human illness in Canada, the literature suggested that a particular strain S. Kentucky ST198 found to be resistant to ciprofloxacin [55] has spread internationally. Poultry has been identified as a major reservoir of this strain [55]. It is important to monitor the prevalence of this serovar as it has been shown to exhibit resistance to tetracycline and aminoglycoside [24]. In this present study, the *S. Kentucky* isolates appear to be the driver for the high-level resistance to tetracycline, as visualized in the heatmaps (thus the decreased occurrence of susceptible isolates). The heatmaps also proved to be a useful tool in illustrating the regional similarities of tetracycline resistance in *S. Kentucky*. Of the top five serovars detected in this current analysis, four were identified by NESP in human clinical isolates (*S. Heidelberg, S. Enteritidis, S. Typhimurium,* and *S. Infantis*) [6]. Similar to the current study, a previous broiler chicken study found *S. Kentucky* (35%) to be the dominating serovar between 2013 and 2018 [56]. Caffrey et al., [56] also reported the same common serovars (*S. Heidelberg, S. Enteritidis, S. Typhimurium,* and *S. Infantis*) identified in this current study. These serovars are likely adapted to the chicken environment and could persist for a long time as they are known to have plasmids that enable efficient colonization in poultry [57]. The heatmaps also identified a cluster of MDR *S. Kentucky* in Western Canada. These isolates observed in farm and slaughtered broiler chickens were resistant to Category I antimicrobials such as amoxicillin-clavulanic acid, ceftriaxone, non-susceptible to ciprofloxacin, and resistant to ampicillin and nalidixic acid, but not tetracycline [4]. Whether this observation is associated with the emergence of a single or more new plasmids [58], particular choice of antimicrobials for disease therapy, or both, in broiler chickens in Western Canada warrants further investigation.

Finally, the frequency of recovery of *Campylobacter* exceeded 70% and three times higher in broiler breeders, than in broiler chickens on farms or at slaughter. This might be explained by the longer lifespan of breeders given that increased odds of *Campylobacter* colonization have been positively correlated with age at slaughter [59]. Although there was a lower *Salmonella* prevalence in breeders compared to broilers on farms, the risk of vertical transmission of certain *Salmonella* serovars/MDR *Salmonella* to progeny flocks is highly likely due to the multiplier nature of broiler chicken production and the opportunities for cross-contamination of the products in relevant production facilities (e.g., hatcheries, farms, slaughter plants, grocery stores) [60]. As such the broiler breeder industry has put in place various preventative measures such as enhanced biosecurity and vaccination programs which might explain the observed lower *Salmonella* prevalence at that level.

This study found *E. coli, Salmonella* and *Campylobacter* in broiler breeders that were resistant to antimicrobials at levels lower (in the case of *E. coli*) or higher (in the case of *Salmonella*) than those found in farm and slaughtered broiler chickens. The AMU exposures (i.e., single or multiple active ingredient exposures during the growing period) and shorter life span of broilers (i.e., reversal to susceptible population takes time) may explain the higher levels of resistance to most antimicrobials in *E. coli* compared to broiler breeders (raised for approximately 60 weeks). In *Salmonella*, vertical transmission has been described in the literature and the persistency of certain serovars, notably, *S. Kentucky* in breeders which are also tetracycline resistant may have contributed to the levels of AMR in *Salmonella* in farm and slaughtered broilers.

This study has certain limitations, firstly, for tracking the foodborne bacteria and their AMR profiles (percentage of resistance and phenotypic patterns) across the food production to fork continuum was not possible due to logistical and data confidentiality concerns. The Canadian poultry production structure under the current supply management system is not vertically integrated where, like in the United States and many countries, breeder flocks to

the slaughter stage is owned by a single company [10,55]. Such vertical integration makes it possible to track the source of resistant microorganisms and relate it to their health records. Secondly, there was heterogeneity in terms of sample matrix used. The sample matrix used for the farm component was pooled floor fecal samples likely to have higher concentration of bacteria and involving several bird feces, whereas the slaughter plant component used cecal samples from individual birds

## Conclusions

This study offers a national perspective of AMR in the broiler chicken food production continuum. The data indicated the persistency of bacteria resistant to antimicrobials categorized by the WHO as HPCIA's and by Health Canada's VDD of very high importance (fluoroquinolones and 3rd generation cephalosporins), with levels of resistance ranging from very low to high. However, MDR *Salmonella* and MDR *Campylobacter* remained low, and resistance to colistin and meropenem was rarely or not detected. The information from this study concerning the frequency of foodborne pathogens (*Salmonella* and *Campylobacter*), levels of MDR, and levels of resistance to WHO's HPCIA, could be used for foodborne pathogen and the risk management of both vertically and horizontally transmitted AMR bacteria and resistance genes. These approaches could include further enhancements of *Salmonella* vaccination schemes in Canadian broiler breeders (efficacious against the common serovars identified), foodborne pathogen (*Salmonella* and *Campylobacter*) reduction, and AMU stewardship. Implementation of these risk mitigation steps in a harmonized manner across the food production to fork continuum complemented by ongoing foodborne pathogen, AMR, and AMU surveillance is essential in the containment of AMR in broiler chicken meat production and microbial safety of products available to Canadian consumers. More importantly, further analysis of potential AMU-AMR associations in breeders and then, AMU-AMR in broilers and molecular analysis of the isolates from broiler breeders and broilers (farm, slaughter plant) would provide a better understanding of the ecology of AMR in the broader broiler chicken sector and trace potential sources of MDR organisms.

## Acknowledgments

The authors acknowledge Dr. Kathleen Long (Ontario) and Dr. Benoit Lanthier (Québec) for their support in funding acquisition. We thank the processing plant quality assurance staff and the sentinel farm veterinarians and their producers for their voluntary participation that enabled data and sample collection.

## Author contributions

**Conceptualization:** Martine Boulianne, Teresa Cereno, Anne E. Deckert, Agnes Agunos.

**Data curation:** Sarah Hill, Anne E. Deckert, Sheryl P. Gow, Kathryn McDonald.

**Formal analysis:** Hiddecel Medrano, Sarah Hill, Sheryl P. Gow, Kathryn McDonald, Agnes Agunos.

**Funding acquisition:** Martine Boulianne, Teresa Cereno, Anne E. Deckert, Richard J. Reid-Smith, Agnes Agunos.

**Investigation:** Martine Boulianne, Anne E. Deckert, Audrey Charlebois, Agnes Agunos.

**Methodology:** Martine Boulianne, Anne E. Deckert, Audrey Charlebois, Sheryl P. Gow, Agnes Agunos.

**Resources:** Martine Boulianne, Richard J. Reid-Smith.

**Supervision:** Richard J. Reid-Smith.

**Validation:** Hiddecel Medrano, Sarah Hill, Anne E. Deckert, Sheryl P. Gow, Kathryn McDonald.

**Visualization:** Hiddecel Medrano, Sarah Hill.

**Writing – original draft:** Hiddecel Medrano, Agnes Agunos.

**Writing – review & editing:** Hiddecel Medrano, Sarah Hill, Martine Boulianne, Teresa Cereno, Anne E. Deckert, Audrey Charlebois, Sheryl P. Gow, Kathryn McDonald, Agnes Agunos.

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
