## [Decision Letter · Decision Letter 0]

3 Dec 2024

PONE-D-24-52074Widespread dissemination of Salmonella, Escherichia coli and Campylobacter resistant to medically important antimicrobials in the poultry production continuum in CanadaPLOS ONE

Dear Dr. Agunos,

Thank you for submitting your manuscript to PLOS ONE. After careful consideration, we feel that it has merit but does not fully meet PLOS ONE’s publication criteria as it currently stands. Therefore, we invite you to submit a revised version of the manuscript that addresses the points raised during the review process.

We look forward to receiving your revised manuscript.

Kind regards,

Mabel Kamweli Aworh, DVM, MPH, PhD. FCVSN

Academic Editor

PLOS ONE

Additional Editor Comments:

In addition to addressing the reviewers' comments please fix the following issues;

1. When discussing your results, avoid repeating the findings already presented in the Results section. Instead, focus on providing explanations for the observed outcomes, highlighting their significance and implications. Compare your findings with those of similar studies to situate your work within the broader research context. Avoid citing figures or tables in the Discussion section; these should be addressed in detail within the Results section. For example, in lines 509-514, 522-524, 570, 601, 626-630, please delete the specific results, just provide an interpretation of the results and implications. Remove figure citation in line 622, please.

2. Please address the disparity in referencing style in line 629.

3. Please highlight some key limitations of this study in the last paragraph of the discussion section.

4. Ensure that the conclusions are presented as a separate section, distinct from the Discussion. In the Conclusion section, succinctly summarize the key findings of your study and their broader implications. In the final paragraph, provide actionable recommendations or outline potential future research directions based on your study findings to guide further exploration in the field.

Reviewers' comments:

Reviewer's Responses to Questions

**Comments to the Author**

1. Is the manuscript technically sound, and do the data support the conclusions?

Reviewer #1: Yes

Reviewer #2: Yes

Reviewer #3: Yes

2. Has the statistical analysis been performed appropriately and rigorously? 

Reviewer #1: Yes

Reviewer #2: Yes

Reviewer #3: Yes

3. Have the authors made all data underlying the findings in their manuscript fully available?

Reviewer #1: Yes

Reviewer #2: Yes

Reviewer #3: Yes

4. Is the manuscript presented in an intelligible fashion and written in standard English?

Reviewer #1: Yes

Reviewer #2: Yes

Reviewer #3: Yes

5. Review Comments to the Author

Reviewer #1: This is a credible work in establishing the AMR and AMU in poultry production in Canada. However, see the minor revisions included. However, I have a few minor suggestions for improvement. Please see the revisions I’ve noted below.

General

The in-text citation is not consistent: PLOS one policy states, “In the text, cite the reference number in square brackets (e.g., “We used the techniques developed by our colleagues [19] to analyze the data”). PLOS uses the numbered citation (citation-sequence) method and first six authors, et al.”

See PLOS One policy regarding nomenclature “Write in italics (e.g., Homo sapiens). Write out in full the genus and species, both in the title of the manuscript and at the first mention of an organism in a paper. After first mention, the first letter of the genus name followed by the full species name may be used (e.g., H. sapiens). Line 390, Line 409-417 and other sections.

Methods

Sample Collection: The methods describe the sampling process from various bird groups, but the exact number of farms involved in the study is not specified. Could you clarify how many farms were included in the sampling for each group of birds (broiler breeders, farm broiler chickens, and slaughtered broiler chickens)? Additionally, it would be helpful to know if the number of farms varied by sampling group or over the study timeframe." Regarding confidentiality for participating establishments, designating Farm 1, 2…. That is enough.

Results/Discussion

You stated that a χ2 test was performed to identify significant mean group differences in % of resistance among study samples in the methods section, but nothing was mentioned about this in the result. Even if the results were not significant, they should be included in the results.

Line 360: The manuscript states in the Results section, "Molecular analysis (not shown) indicated that these isolates are mobilized colistin resistant-1 (mcr-1) gene negative." PLOS ONE requires all data supporting the study's findings to be accessible. I recommend including the molecular analysis results in the text or supplementary material to ensure transparency and reproducibility or this statement should be omitted.

For heat maps, on the y-axis, kindly include the species or names of the bacteria at each node or a color key to show which organisms/species are at each node.

Fig 5. Number of Salmonella serovars recovered from chickens,, but there is no mention of the serovars 'I:8,20:-:26' and 'i:8,20:i:-' in the result and discussion section. Could you clarify whether these serovars have yet to be identified or if they are new serovars that have not been previously reported in the study? Further details on their classification or status would help to better interpret the results."

Reviewer #2: Comments

The Canadian Integrated Program for Antimicrobial Resistance Surveillance (CIPARS) monitors Escherichia coli, Salmonella and Campylobacter and their resistance to antimicrobials in broiler chickens at the farm and slaughter plant levels. The study assessed the levels of the aforementioned bacteria and their antimicrobial resistance in view of the fact that resistance genes can be transmitted vertically from parents to their offspring. It was found that broiler breeders carry foodborne bacteria and exhibit resistance to antimicrobials.

Recommendation

The manuscript is well written. The theme of the research, drug resistance, is critical to health delivery, be they human beings or lower animals 9vetrinary). The paper is therefore recommended for publication.

Reviewer #3: The study provides a good overview of microbial presence and their resistance to antimicrobial agents. The authors did a commendable job in defining the problem and drawing clear conclusions. Additionally, I appreciate the suggestion for further investigation to understand whether there are any genetic explanations behind the increase in resistance.

Given their work, I would recommend the paper for publication, but a few adjustments should be made before making it publicly available.

Although the results are understandable, the sentence in lines 38-40 can be confusing as it may imply that the susceptibility was for all microbes, which could affect the understanding of the next sentence (lines 40-42). Therefore, the authors need to specify that the highest susceptibility is from E. coli in the sentence: “Broiler breeders had the highest susceptibility (54%) to all antimicrobials compared to farm (36%) and slaughtered (35%) broiler chickens.”

In line 144, “a small proportion of flocks in two provinces were sampled in early 2022,” it would be appreciated if the authors specify that proportion and which provinces they are referring to.

Unless I missed it in the paragraphs, is there any reason why the sample size (N) of slaughtered broiler chickens is different across the microbial agents (Table 1)?

Line 291: Add CI after 95%. “Of the broiler breeder isolates, 7.5% (95% 6.3-8.9%) were resistant to three or more antimicrobial classes.”

Another minor change should be made to the tables to ensure consistency with the colors. Also, ensure consistency when referring to figures (e.g., Fig vs. Figure).

Finally, although it may not be technically required, reporting the p-values or using FDR, and conducting a multiple comparison analysis would be appreciated to check differences among all groups (breeders, farm, and slaughtered chickens) rather than limiting the comparisons to breeders only.

6. PLOS authors have the option to publish the peer review history of their article (what does this mean? ). If published, this will include your full peer review and any attached files.

**Do you want your identity to be public for this peer review?** For information about this choice, including consent withdrawal, please see our Privacy Policy .

Reviewer #1: **Yes: ** Damilola Odumade

Reviewer #2: **Yes: ** Taiwo Akindahunsi

Reviewer #3: No

---

## [Author Response · Author response to Decision Letter 1]

3 Jan 2025

Responses to PlosOne Editors and Reviewers:

Dear Sir/Madam,

We thank you for the opportunity to revise our manuscript for consideration to publish in Plos One. We have addressed your comments and recommendations to improve the manuscript.

We look forward to hearing from you regarding the revised version. Please do not hesitate to contact me if you have any further questions.

Sincerely,

Agnes Agunos, DVM, MSc, DipACPV

Canadian Integrated Program for Antimicrobial Resistance Surveillance (CIPARS)

Public Health Agency of Canada

Tel: 519-826-2351, Fax: 519-826-2255, Email: agnes.agunos@phac-aspc.gc.ca

Authors’ response: Thank you for providing additional guidance. Style requirements and file naming were adjusted accordingly.

Author’s response: Ethics statement was added in the Methods (last section under “Farmer informed consent”.

Additional Editor Comments:

In addition to addressing the reviewers' comments please fix the following issues;

1. When discussing your results, avoid repeating the findings already presented in the Results section. Instead, focus on providing explanations for the observed outcomes, highlighting their significance and implications. Compare your findings with those of similar studies to situate your work within the broader research context. Avoid citing figures or tables in the Discussion section; these should be addressed in detail within the Results section. For example, in lines 509-514, 522-524, 570, 601, 626-630, please delete the specific results, just provide an interpretation of the results and implications. Remove figure citation in line 622, please.

Authors’ response: Thank you very much for these comments; statements were revised accordingly to avoid duplication with the results section and those already provided in the figures and tables. Other areas were checked if the implications of the results were missing.

2. Please address the disparity in referencing style in line 629.

Authors’ response: the reference was added to the list and the in-text citation was adjusted accordingly.

3. Please highlight some key limitations of this study in the last paragraph of the discussion section.

Author’s response: the key limitations of this study are highlighted in a paragraph, starting on Line 712.

4. Ensure that the conclusions are presented as a separate section, distinct from the Discussion. In the Conclusion section, succinctly summarize the key findings of your study and their broader implications. In the final paragraph, provide actionable recommendations or outline potential future research directions based on your study findings to guide further exploration in the field.

Author’s response: we thank you for these comments. The Conclusion section now highlights what the study could offer or its contribution to knowledge concerning the ecology of AMR in the broiler chicken meat sector.

Reviewers' comments:

Reviewer's Responses to Questions

Comments to the Author

1. Is the manuscript technically sound, and do the data support the conclusions?

Reviewer #1: Yes

Reviewer #2: Yes

Reviewer #3: Yes

2. Has the statistical analysis been performed appropriately and rigorously?

Reviewer #1: Yes

Reviewer #2: Yes

Reviewer #3: Yes

3. Have the authors made all data underlying the findings in their manuscript fully available?

Reviewer #1: Yes

Reviewer #2: Yes

Reviewer #3: Yes

4. Is the manuscript presented in an intelligible fashion and written in standard English?

Reviewer #1: Yes

Reviewer #2: Yes

Reviewer #3: Yes

5. Review Comments to the Author

Reviewer #1: This is a credible work in establishing the AMR and AMU in poultry production in Canada. However, see the minor revisions included. However, I have a few minor suggestions for improvement. Please see the revisions I’ve noted below.

Authors’ response: we thank the reviewer for this positive feedback.

General

The in-text citation is not consistent: PLOS one policy states, “In the text, cite the reference number in square brackets (e.g., “We used the techniques developed by our colleagues [19] to analyze the data”). PLOS uses the numbered citation (citation-sequence) method and first six authors, et al.”

See PLOS One policy regarding nomenclature “Write in italics (e.g., Homo sapiens). Write out in full the genus and species, both in the title of the manuscript and at the first mention of an organism in a paper. After first mention, the first letter of the genus name followed by the full species name may be used (e.g., H. sapiens). Line 390, Line 409-417 and other sections.

Authors’ response: well noted and all in text citations were universally checked and changed into square brackets, following PLOS ONE recommendations.

Methods

Sample Collection: The methods describe the sampling process from various bird groups, but the exact number of farms involved in the study is not specified. Could you clarify how many farms were included in the sampling for each group of birds (broiler breeders, farm broiler chickens, and slaughtered broiler chickens)? Additionally, it would be helpful to know if the number of farms varied by sampling group or over the study timeframe." Regarding confidentiality for participating establishments, designating Farm 1, 2…. That is enough.

Authors’ response: well noted and average flock sizes or average slaughter plant participating in the programmes were added. Please see lines #149-151, 155 and 160-162.

Results/Discussion

You stated that a χ2 test was performed to identify significant mean group differences in % of resistance among study samples in the methods section, but nothing was mentioned about this in the result. Even if the results were not significant, they should be included in the results.

Author’s response. We thank you for this comment. This was one of the exploratory analytic methods for comparing between production stages. We retained the logistic regression results which were appropriate for the data as the methodology enabled adjustments for clustering to account for multiple isolates collected per flock/farm or batch of birds at slaughter.

Line 360: The manuscript states in the Results section, "Molecular analysis (not shown) indicated that these isolates are mobilized colistin resistant-1 (mcr-1) gene negative." PLOS ONE requires all data supporting the study's findings to be accessible. I recommend including the molecular analysis results in the text or supplementary material to ensure transparency and reproducibility or this statement should be omitted.

Authors’ response: Thank you for this comment, the statement about the molecular analysis of mcr-1 was deleted.

For heat maps, on the y-axis, kindly include the species or names of the bacteria at each node or a color key to show which organisms/species are at each node.

Authors’ response. We thank you for this comment. The meat maps were used to summarize the relationships of the isolates based on their AMR profiles (not to be confused with phylogenetic trees showing relationship of a specific attribute of the isolates). The description of the heat map was included in the methods on lines 241 to 256. The heat maps (3 total) in the manuscripts correspond to one bacteria each and for the Salmonella, it was only S. Kentucky, the predominant serovar that was presented in the heat map for this bacteria. As explained, the y-axis on the right are the isolate identifiers and are re-ordered based on hierarchal clustering.

Fig 5. Number of Salmonella serovars recovered from chickens,, but there is no mention of the serovars 'I:8,20:-:26' and 'i:8,20:i:-' in the result and discussion section. Could you clarify whether these serovars have yet to be identified or if they are new serovars that have not been previously reported in the study? Further details on their classification or status would help to better interpret the results."

Authors’ response: thank you for these comments. We have revised the figures to reflect the diversity of serovars recovered and their proportions. There is a category known as less-frequently occurring serovars (less than or equal to 1% of the total isolates). We presented only those that were frequently occurring.

Reviewer #2: Comments

The Canadian Integrated Program for Antimicrobial Resistance Surveillance (CIPARS) monitors Escherichia coli, Salmonella and Campylobacter and their resistance to antimicrobials in broiler chickens at the farm and slaughter plant levels. The study assessed the levels of the aforementioned bacteria and their antimicrobial resistance in view of the fact that resistance genes can be transmitted vertically from parents to their offspring. It was found that broiler breeders carry foodborne bacteria and exhibit resistance to antimicrobials.

Recommendation

The manuscript is well written. The theme of the research, drug resistance, is critical to health delivery, be they human beings or lower animals 9vetrinary). The paper is therefore recommended for publication.

Author’s response: thank you for this very positive and encouraging comment.

Reviewer #3: The study provides a good overview of microbial presence and their resistance to antimicrobial agents. The authors did a commendable job in defining the problem and drawing clear conclusions. Additionally, I appreciate the suggestion for further investigation to understand whether there are any genetic explanations behind the increase in resistance.

Given their work, I would recommend the paper for publication, but a few adjustments should be made before making it publicly available.

Although the results are understandable, the sentence in lines 38-40 can be confusing as it may imply that the susceptibility was for all microbes, which could affect the understanding of the next sentence (lines 40-42). Therefore, the authors need to specify that the highest susceptibility is from E. coli in the sentence: “Broiler breeders had the highest susceptibility (54%) to all antimicrobials compared to farm (36%) and slaughtered (35%) broiler chickens.”

Authors response: thank you for this comment, the statements were revised accordingly.

In line 144, “a small proportion of flocks in two provinces were sampled in early 2022,” it would be appreciated if the authors specify that proportion and which provinces they are referring to.

Authors’ response: thank you for this comment. The text was adjusted accordingly.

Unless I missed it in the paragraphs, is there any reason why the sample size (N) of slaughtered broiler chickens is different across the microbial agents (Table 1)?

Authors’ response: thank you for this comment. We have provided a footnote in Table about the sample sizes concerning the slaughtered broiler chicken samples. “A total of 3,089 samples were submitted during the study timeframe. Final sample size varied depending on the organism: E. coli (1 in 4 samples were tested), Salmonella (13 samples were unfit for testing or lost specimen) and Campylobacter (24 samples unfit for testing or lost specimen).”

Line 291: Add CI after 95%. “Of the broiler breeder isolates, 7.5% (95% 6.3-8.9%) were resistant to three or more antimicrobial classes.”

Authors’ response. Thank you for this comment – it was adjusted accordingly.

Another minor change should be made to the tables to ensure consistency with the colors. Also, ensure consistency when referring to figures (e.g., Fig vs. Figure).

Authors’ response. Well noted and adjusted accordingly.

Finally, although it may not be technically required, reporting the p-values or using FDR, and conducting a multiple comparison analysis would be appreciated to check differences among all groups (breeders, farm, and slaughtered chickens) rather than limiting the comparisons to breeders only.

Authors’ response: the authors thank the reviewer for this comment. A more detailed analysis or modelling such as time series is underway to understand the relationship between AMR in broiler breeders and AMR in the progeny flocks. For example, if the AMR in breeders during their laying phase (or peak of laying phase) could be predict the AMR levels in progeny flocks (estimated hatch and growing period – 3 months after). The Salmonella and E. coli isolates will also be further characterized using WGS to complement the heat maps.

6. PLOS authors have the option to publish the peer review history of their article (what does this mean?). If published, this will include your full peer review and any attached files.

Do you want your identity to be public for this peer review? For information about this choice, including consent withdrawal, please see our Privacy Policy.

Reviewer #1: Yes: Damilola Odumade

Reviewer #2: Yes: Taiwo Akindahunsi

Reviewer #3: No

[NOTE: If reviewer comments were submitted as an attachment file, they will be attached to this email and accessible via the submission site. Please log into your account, locate the manuscript record, and

---

## [Decision Letter · Decision Letter 1]

28 Jan 2025

Widespread dissemination of Salmonella, Escherichia coli and Campylobacter resistant to medically important antimicrobials in the poultry production continuum in Canada

PONE-D-24-52074R1

Dear Dr. Agunos

We’re pleased to inform you that your manuscript has been judged scientifically suitable for publication and will be formally accepted for publication once it meets all outstanding technical requirements.

Kind regards,

Mabel Kamweli Aworh, DVM, MPH, PhD. FCVSN

Academic Editor

PLOS ONE

Additional Editor Comments (optional):

Reviewers' comments:

Reviewer's Responses to Questions

**Comments to the Author**

1. If the authors have adequately addressed your comments raised in a previous round of review and you feel that this manuscript is now acceptable for publication, you may indicate that here to bypass the “Comments to the Author” section, enter your conflict of interest statement in the “Confidential to Editor” section, and submit your "Accept" recommendation.

Reviewer #1: All comments have been addressed

Reviewer #3: All comments have been addressed

2. Is the manuscript technically sound, and do the data support the conclusions?

Reviewer #1: Yes

Reviewer #3: (No Response)

3. Has the statistical analysis been performed appropriately and rigorously? 

Reviewer #1: Yes

Reviewer #3: (No Response)

4. Have the authors made all data underlying the findings in their manuscript fully available?

Reviewer #1: Yes

Reviewer #3: (No Response)

5. Is the manuscript presented in an intelligible fashion and written in standard English?

Reviewer #1: Yes

Reviewer #3: (No Response)

6. Review Comments to the Author

Reviewer #1: Thank you for your contribution to the science of antimicrobial resistance. This is a well-written work.

Reviewer #3: (No Response)

7. PLOS authors have the option to publish the peer review history of their article (what does this mean? ). If published, this will include your full peer review and any attached files.

**Do you want your identity to be public for this peer review?** For information about this choice, including consent withdrawal, please see our Privacy Policy .

Reviewer #1: **Yes: ** Damilola Odumade

Reviewer #3: No

---

## [Editor Report · Acceptance letter]

PONE-D-24-52074R1

PLOS ONE

Dear Dr. Agunos,

I'm pleased to inform you that your manuscript has been deemed suitable for publication in PLOS ONE. Congratulations! Your manuscript is now being handed over to our production team.

Kind regards,

on behalf of

Dr. Mabel Kamweli Aworh

Academic Editor

PLOS ONE